# What is Missing? Explaining Neurons Activated by Absent Concepts

## Abstract

Explainable artificial intelligence (XAI) aims to provide human-interpretable insights into the behavior of deep neural networks (DNNs), typically by estimating a simplified causal structure of the model. In existing work, this causal structure often includes relationships where the presence of an input pattern or latent feature is associated with a strong activation of a neuron. For example, attribution methods primarily identify input pixels that contribute most to a prediction, and feature visualization methods reveal inputs that cause high activation of a target neuron – both implicitly assuming that neurons encode the presence of concepts. However, a largely overlooked type of causal relationship is that of *encoded absences*, where the absence of a concept increases activation, or vice versa, the presence of a concept inhibits activation. In this work, we show that such inhibitory relationships are common and that mainstream XAI methods struggle to reveal them when applied in their standard form. To address this, we propose two extensions to attribution and feature visualization techniques that uncover encoded absences. Across experiments, we show how mainstream XAI methods can be used to reveal and explain encoded absences, how ImageNet models exploit them, and that debiasing can be improved when considering them.

## 1 Introduction

Two of the arguably most important methods in explainable artificial intelligence (XAI) – namely, attribution and feature visualization techniques – primarily associate the activation of a neuron with the *presence* of specific concepts. For instance, attribution methods highlight which features present in the input have been relevant to activate the neuron corresponding to the target class, and feature visualization methods find input patterns whose presence maximizes a neuron's activation.

However, in biological neural networks, presences are only one side of the story. Equally important are *absences*, which often serve as powerful reasoning cues. *E.g.*, in clinical diagnosis, humans may pay closer attention to the absence of specific symptoms than to the proper functioning of dozens of physiological processes. Likewise, the Hassenstein–Reichardt model (Egelhaaf et al., 1989) describes neurons in the *Drosophila melanogaster* that are activated by the presence of rightward motion in combination with the absence of leftward motion, enabling the fly to distinguish rightward motion from predators whose looming movement produces motion in multiple directions (see Fig. 1, left).

Interestingly, work on explaining neurons in deep neural networks (DNNs) through logical compositions (Mu & Andreas, 2020) has revealed related phenomena in artificial neural networks, where a neuron can encode the logical NOT ($\neg$) of a concept – an idea connected to, though fundamentally distinct from, the notion of representing a concept's absence that we consider and define here (see Section 3). Moreover, this insight has not been integrated into "mainstream" XAI, such as attribution and feature visualization methods, likely due to the algorithmic complexity of existing approaches, which require annotated concept datasets and rely on expensive optimization over logical compositions.

In this work, we close this gap by showing how standard attribution and feature visualization approaches can be used for illuminating encoded absences in DNNs – *i.e.*, features *not* in the input but still causally linked to the prediction (exemplified with image classification models). By doing so, we show that absences are especially relevant for fine-grained classification, where subtle differences matter: distinguishing an Irish Setter from a Sussex Spaniel benefits not only from detecting Setter-specific features but also from confirming the *absence* of Spaniel-specific ones (see Fig. 1, right).

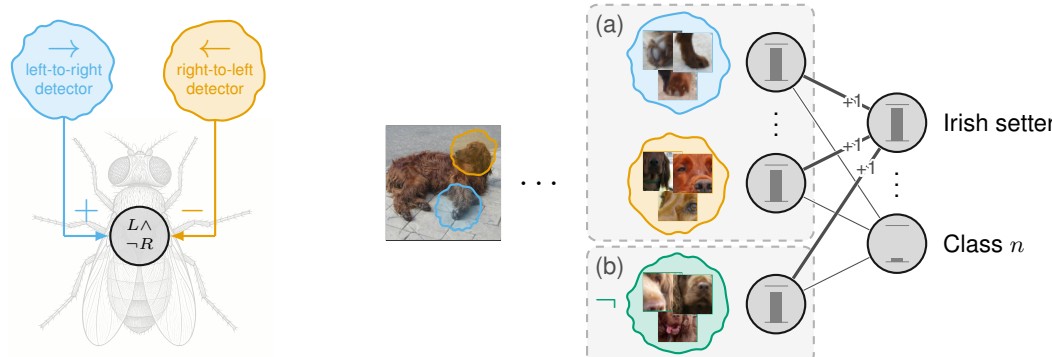

Figure 1: *(left)* **Simplified illustration of the Hassenstein-Reichardt detector in *Drosophila*.** The activation of two subunits – encoding right-to-left ($R$) and left-to-right ($L$) movements – is subtracted. The output neuron encodes the *presence* of left-to-right movements while encoding the *absence* of right-to-left movements ($L \wedge \neg R$). *(right)* **Illustration of an image classification model with an encoded absence.** *(a)* The model detects concepts present in the input image that are prototypical for the target class (*e.g.*, the snout and feet). *(b)* The model can additionally encode the absence of snouts from other dog species to enhance evidence for the "Irish setter" class.

Further, we show how our proposed modifications can be used for debiasing models based on absences. More specifically: *(i)* We formally define encoded absences and show that they represent a largely overlooked causal relationship for understanding DNNs. *(ii)* We illustrate how DNNs encode such absences on a mechanistic level. *(iii)* We show why existing mainstream explanation methods – namely, attribution methods, feature visualization techniques, and counterfactual explanations – struggle to capture them in their standard form. *(iv)* We show how attribution and feature visualization methods can be utilized to explain encoded absences. *(v)* We empirically verify our theoretical findings and study whether and how absences are used in image classification, and how they can contribute to debiasing.

## 2 ENCODING THE ABSENCE OF LATENT FEATURES

Although previous work in XAI for image classification has shown that neurons can participate in logical expressions involving a NOT operator (Mu & Andreas, 2020), these approaches do not explicitly consider inhibitory signals, and consequently, do not capture the broader notion of encoded absences we formalize here. To highlight this distinction, we introduce a causal formulation of encoded absences and outline how such inhibitory relationships can arise in neural representations. Beyond separating our approach from related work, this highlights why encoded absences matter for XAI, clarifies how attribution and feature-visualization methods can reveal them (*cf.* Section 4), and provides mechanistic insight into their implementation.

### 2.1 A CAUSAL PERSPECTIVE: WHY ENCODING ABSENCES IS INTERESTING

The goal of XAI can be reframed as finding a *simplified* approximation of the DNN's underlying causal structure (Hesse et al., 2023; Carloni et al., 2025). While the *true* causal structure is embodied by the DNN itself, its complexity typically exceeds human understanding. Thus, a "simplified" structure refers to one that enables a human to understand the model sufficiently to answer task-specific questions of interest. Since the appropriate level of simplification depends on both the user and the task (Tomsett et al., 2019), a wide range of causal abstractions could be relevant – and should be explored within XAI research. Formally, a feed-forward DNN $f \colon \mathbb{R}^n \mapsto \mathbb{R}$ can be expressed as a structural causal model (SCM) $\mathfrak{C} \coloneqq (\mathbf{S})$ (Peters et al., 2017) with structural assignments $\mathbf{S}$ defining each intermediate representation as a deterministic function of its parents, *i.e.*, $z^{(1)} \coloneqq f^{(1)}(x)$, $z^{(2)} \coloneqq f^{(2)}(z^{(1)})$, ..., $y \coloneqq f^{(n)}(z^{(n-1)})$, where $x$ is the input; the noise variables usually found in SCMs are set to zero for simplicity. In XAI, we seek a simplified SCM $\mathfrak{C}'$ that approximates the original SCM $\mathfrak{C}$ in a way that preserves task-relevant causal relationships while improving human interpretability (Hesse et al., 2023; Carloni et al., 2025). *E.g.*, in the case of a

simple gradient-based attribution method (Simonyan et al., 2014), $\mathfrak{C}'$ would be a linear approximation of the structural assignment $y := f(x)$, where each feature $x_i$ is associated with a causal influence estimated by $\frac{\partial f(x)}{\partial x_i}$ (see Appendix A.1 for feature visualization and counterfactual explanations).

A less studied causal relationship in XAI involves concepts whose absence causes higher activations, or vice versa, whose presence suppresses the activation of a specific internal neuron $z_j$ or output $y$. Formally, let $C_{\hat{x}} \in \{0, 1\}$ denote a binary variable indicating whether the concept $\hat{x}$ is present in the input $[x, C_{\hat{x}}]$; then such an inhibitory relationship holds whenever $f\big(do(x := [x, C_{\hat{x}} = 0])\big) > f\big(do(x := [x, C_{\hat{x}} = 1])\big)$, i.e., introducing the concept via $do(x := [x, C_{\hat{x}} = 1])$ decreases the activation. Intuitively, such interventions reveal patterns that actively suppress a neuron's activation, akin to the illustrative example of the Hassenstein–Reichardt detector, where the opposite motion direction inhibits the response. We outline how to uncover such a causal relationship in Section 4.

**Definition 2.1 (Encoded Absence)** *If there exists an input pattern/concept $\hat{x}$ whose presence causes the activation of a neuron $z_j$ in layer $l$ to decrease,* i.e., $f_j^{(l)}([x, C_{\hat{x}} = 1]) < f_j^{(l)}([x, C_{\hat{x}} = 0])$, *we say that the neuron $z_j$ encodes the* absence *of said feature $\hat{x}$ in the input context of $x$.*

## 2.2 A MECHANISTIC PERSPECTIVE: HOW TO ENCODE ABSENCES

Having established that neurons encoding the *absence* of a latent feature can contribute to more complete explanations, we now present a constructive existence proof demonstrating that neural networks are capable of implementing such neurons.[1]

**Proposition 2.2** *Neural networks can implement neurons $z_j$ that encode the absence of a latent feature $\hat{x}$.*

For our construction, we assume that in layer $l-1$, each neuron encodes the *presence* of one or multiple latent features $\{\hat{x}, ...\}$; if a neuron in $l-1$ already encoded the *absence* of a feature, the proof would be trivially complete. Furthermore, we assume unnormalized activations and that the model employs a ReLU activation function (Fukushima, 1969) as found in many image classification models – these assumptions are introduced solely for simplification; without them, even more strategies for encoding absences are possible, as outlined in Appendix A.2.

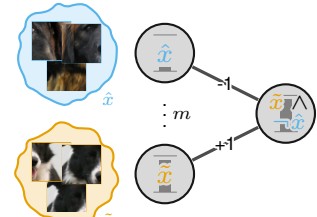

Figure 2: **A mechanistic process to encode the absence of a feature.** A neuron encoding the *absence* of latent feature $\hat{x}$ (*i.e.* $\neg\hat{x}$) can be implemented by having a negative connection to a neuron encoding $\hat{x}$ and a positive potential through, *e.g.*, another activating concept $\tilde{x}$ (*i.e.*, the output encodes $\tilde{x} \wedge \neg\hat{x}$).

A simple way to construct a neuron $z_j$ in layer $l$ that encodes the absence of the latent feature $\hat{x}$ encoded in layer $l-1$ involves two components: *(i)* negative weights connecting neurons in $l-1$ that encode the *presence* of $\hat{x}$ to $z_j$, and *(ii)* a source of positive potential to ensure that $z_j$ is activated when $\hat{x}$ is *absent*. When both conditions are met, $z_j$ will produce a high activation if $\hat{x}$ is *absent*, and a low activation if $\hat{x}$ is *present* – effectively encoding the *absence* of $\hat{x}$. This construction is illustrated in Fig. 2. The positive potential can be supplied, *e.g.*, by using the activation of another latent feature $\tilde{x}$ in $l-1$. As a result, $z_j$ jointly encodes the *presence* of $\tilde{x}$ and the *absence* of $\hat{x}$. In Appendix A.2, we provide additional mechanisms to encode absences and extend the idea to polysemantic neurons, respectively, concepts that lie on arbitrary feature space directions (Elhage et al., 2022).

## 3 RELATED WORK

**Explaining encoded absences.** Related ideas to encoded absences have been studied in prior XAI research. For instance, explanations based on logical compositions revealed that neurons can participate in expressions that include a logical NOT. Mu & Andreas (2020) use Network Dissection (Bau et al., 2017) followed by a compositional search to discover logical formulae that approximate neuron behavior. Rosa et al. (2023) extend this idea by accounting for different activation ranges rather than only peak activations. However, these methods come with several limitations: they require an annotated concept dataset, involve computationally expensive searches, and – particularly for ReLU-based models – evaluate logical NOT only on samples where the neuron is positively

---

[1]While previous work has already shown that neural networks are capable of implementing logical NOT operations (Dukor, 2018) – which is similar to encoding absences – we offer a more rigorous proof here.

activated, thereby neglecting a large portion of the data. Most importantly, because these methods rely on intersections of binary masks, a concept is treated as part of a logical NOT whenever it simply fails to activate a neuron. On the other hand, our encoded absences require evidence of active suppression (see Section 5.3 for an experiment demonstrating this). Also loosely related is the notion of *criticism* (Kim et al., 2016), where explanations include samples that are not well captured by class-specific prototypes. While this offers insight into corner cases missed by prototype explanations, the method does not aim to identify *inhibitory* signals or concept-driven suppression, which is the focus of our notion of encoded absence. Oikarinen & Weng (2024) consider the full activation range for explanations; however, for ReLU-based models they operate on post-ReLU activations, which collapse inhibitory signals in the negative range to zero, making them indistinguishable from simply non-activating concepts. Beyond these post-hoc techniques, several works have incorporated encoded absences directly into inherently explainable model architectures. For example, Prabhushankar & AlRegib (2021) and Singh & Yow (2021) design neural networks that explicitly perform negative reasoning, *i.e.*, make predictions by leveraging absences. As such models cannot be applied post hoc to arbitrary networks, we consider them beyond the scope of this work.

Although related ideas exist, encoded absences are largely missing from mainstream XAI, as shown by the three most common XAI methods, which in their standard form struggle to reveal them.

**Attribution methods.** Attribution methods indicate how important each input feature is to the (intermediate) output of a DNN, *e.g.*, by backward propagation of the output signal to the input (Bach et al., 2015), or by evaluating the input gradients (Simonyan et al., 2014; Sundararajan et al., 2017). *Limitation:* By design, attribution methods only identify regions *present* within the input as relevant to the model's output. Thus, explaining that the *absence* of a concept – *i.e.*, a concept that is *not* within the input – was important for the prediction cannot be directly achieved with attribution methods as they are currently used. That said, to some extent, negative attributions (*e.g.*, Lundberg & Lee, 2017) can be interpreted as indicating inhibitory signals, *i.e.*, signals that enhance the activation when the feature is absent, which we exploit in Section 4. However, many methods focus solely on the absolute value of attributions (Simonyan et al., 2014; Srinivas & Fleuret, 2019; Yang et al., 2023), effectively ignoring the sign and thus overlooking whether a feature contributes positively or negatively.

**Feature visualization.** Feature visualization highlights concepts encoded in a DNN neuron (Erhan et al., 2009; Simonyan et al., 2014; Olah et al., 2017), *e.g.*, by finding the inputs that most positively activate a neuron (Olah et al., 2017; Rao et al., 2024; Hesse et al., 2025), or by optimizing the input to maximize the target neuron activation (Simonyan et al., 2014; Olah et al., 2017). *Limitation:* For a neuron encoding the absence of a concept, its maximizing input does not depict the concept itself, but rather visualizations that explicitly exclude it. Consequently, feature visualization methods through maximization cannot faithfully explain a neuron encoding the absence of a concept.

**Counterfactual explanations.** Counterfactual explanations (CEs) highlight why a specific prediction was made instead of another (Goyal et al., 2019; Wang et al., 2023; Guidotti, 2024; Verma et al., 2024), *e.g.*, by finding the patches in two images of two classes that result in the maximum change of the prediction when swapped (Goyal et al., 2019). Although the premise of our paper is based on a counterfactual argument (*cf.* Section 2.1), it is fundamentally different from existing CEs: Our approach does not contrast two specific samples or classes but rather contrasts the activation of a neuron against the entire training distribution. Further, our approach is applied on a neuron level, allowing for a mechanistic understanding, whereas most CEs are applied on a class level. Lastly, we do not require "minimal" interventions as most counterfactual explanations do. *Limitation:* Counterfactual explanations lack a notion of *absence* as established in this work. Returning to the introductory example in Fig. 1 (right), a counterfactual explanation could reveal that the dog snout is the discriminative feature distinguishing Irish Setters from Sussex Spaniels. However, this does not expose whether the model is using the *absence* of Sussex Spaniel snouts, the *presence* of Irish Setter snouts, or both. As a result, existing counterfactual explanations cannot be used to explain neurons encoding the absence of a concept.

## 4 EXPLAINING NEURONS THAT ENCODE THE ABSENCE OF A LATENT FEATURE

Equipped with the notion of *encoded absences* and an understanding of the limitations of existing work, we now show how attribution and feature visualization methods can be utilized to explain neurons that encode absences following Definition 2.1.

**Non-target attribution methods.** As discussed in Section 3, attribution methods typically compute a *targeted* attribution $\mathcal{A}(x, t, f)$ for an input $x$ and target $t$ (usually the prediction $t = f(x)$ or the ground truth). These methods highlight features *present* in $x$ but cannot capture concepts relevant for $t$ that are absent from $x$. To address this, we not only compute the attribution for $x$, but also the attribution $\mathcal{A}(x^{(c \neq t)}, t, f)$ for the class $t$ using inputs $x^{(c \neq t)}$ from other classes (or, more generally, from a diverse set of samples). Intuitively, computing the attributions $\mathcal{A}(x^{(c \neq t)}, t, f)$ for $t$ using a diverse set of inputs ensures that *all* concepts from the training distribution influencing $t$ are considered, including those whose *absence* is informative. In particular, if the model relies on the *absence* of a concept to predict class $t$, there will be cases where the attribution of $t$ is computed for an input in which that concept is *present*. According to Definition 2.1, the *presence* of this concept has an inhibitory effect on the output for $t$, and as a result, the attribution for that concept will be negative. This is different from negative values in the targeted attribution $\mathcal{A}(x, t, f)$, since $x$ may not contain all the relevant concepts and therefore cannot reveal their inhibitory role. We call this approach *non-target attribution* to distinguish it from the commonly used target attribution.

The concrete computation of non-target attributions depends – just as in standard attribution methods – on the task at hand. For example, in our qualitative analyses in Sections 5.1 and 5.2, we compute non-target attributions only for the two samples shown in the visualization (see Appendices B.1 and B.2). In our debiasing experiment in Section 5.4, we instead compute the non-target attribution for every training sample, alongside the target attribution, to incorporate both into an attribution prior (see Appendix B.4). Moreover, since feature attribution methods can produce negative attributions for reasons unrelated to inhibitory evidence, such as gradient shattering (Balduzzi et al., 2017), it is important to note that negative non-target attributions do not, by themselves, guarantee the presence of an encoded absence. In our experiments, however, we observed that this limitation did not meaningfully affect the applications we studied, a finding we further substantiate through a controlled analysis in Appendix B.5.

**Feature visualization through minimization.** As discussed in Section 3, feature visualization through maximization cannot visualize the latent features whose absence is encoded by a neuron, as the inputs that maximally activate the specific neuron contain minimal amounts of features that inhibit activation. To account for this problem, we propose to use *feature visualization through minimization* to find the input $\hat{x}$ that minimizes the activation of a neuron $z_j$ (before the activation function), *i.e.*, $\hat{x} = \arg\min_x z_j(x)$. Intuitively, inputs that lead to strong negative activation highlight patterns that *inhibit* the neuron, revealing the subset of concepts whose *absence* the neuron encodes most strongly.

**Overlap with related work.** Interestingly, from an algorithmic standpoint, our proposed modifications are not entirely unprecedented. For instance, in a targeted FGSM adversarial attack (Goodfellow et al., 2015), the input gradient for a sample is computed with respect to a target class different from the true or predicted class. This can be interpreted as computing a non-target attribution, as outlined above. Similarly, Walter et al. (2025) compute attribution maps for multiple classes on the same input to obtain more class-specific explanations. For feature visualization, Olah et al. (2017) also experimented with inputs that minimize the activation of a target neuron to reveal concepts that activate a neuron to varying degrees. While it is in principle well-known that activations can be maximized or minimized, prior work treated minimization merely as a technical variant; its necessity and semantic role in encoding absences have remained largely unexplored.

That said, while these methods share the same underlying algorithms, the intent and interpretive framing differ fundamentally. To the best of our knowledge, no prior work has linked these modifications to the human-understandable notion of encoding absences, *i.e.*, concepts that are not present in the input but still causally affect the model's prediction. Our contribution lies precisely in formalizing this perspective and highlighting that a complete explanation requires examining both encoded presences and absences. Importantly, our modifications are not meant to replace, but to complement, established attribution and feature visualization methods.

## 5 EXPERIMENTS

We now empirically establish that DNNs can and do encode absent concepts, that common XAI methods struggle with them, and that our proposed modifications can visualize these absences. We further briefly demonstrate how ImageNet-trained models make use of absences and how to debias DNNs relying on absent features. Since our contribution is of a conceptual nature, highlighting the

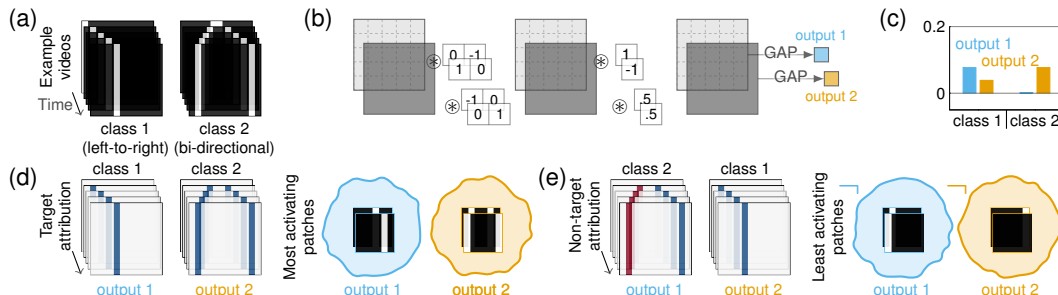

Figure 3: **Hassenstein-Reichardt detector experiment.** *(a)* Two example sequences showing a left-to-right and bi-directional movement. *(b)* A hand-crafted CNN to distinguish left-to-right motion from bi-directional motion. The first layer implements the spatio-temporal comparison of neighboring pixels, the second layer compares motion in opposing directions, followed by global average pooling (GAP). The first output node implements a Hassenstein-Reichardt detector (weights: 1/-1) and the second output averages both directions (weights: 0.5/0.5). *(c)* The outputs of the model for the two example sequences. *(d)* Visualizations of established XAI methods – target attribution and feature visualization for the highest activating patches, each consisting of two consecutive frames as CNN input. Both methods fail to highlight the absence encoded in the first output and thus lack a complete explanation of CNN mechanisms. *(e)* Our proposed non-target attributions and feature visualization via minimization highlight that the first output encodes the absence of right-to-left motion.

relevance of absences for deep neural networks and XAI, we use simple experimental setups to isolate this phenomenon and leave more complex tasks for future work.

## 5.1 EXPLAINING ENCODED ABSENCES IN A HASSENSTEIN-REICHARDT DETECTOR

We first revisit our example from Section 1 – the Hassenstein-Reichardt detector. As input, we generate two video sequences with left-to-right or bi-directional motion as shown in Fig. 3 (a). We manually design a convolutional neural network (CNN) to distinguish these sequences, as shown in Fig. 3 (b). With two consecutive frames as input, the first convolutional layer of the network can extract directional motion features by spatio-temporal comparisons (followed by ReLU activation), similar to the two mirror-symmetric circuits of the Hassenstein-Reichardt detector. For the second layer, the first channel implements a Hassenstein–Reichardt detector by subtracting both directions (kernel of size $(C = 2) \times (H = 1) \times (W = 1)$ and weights of 1 and $-1$), while the second channel performs bi-directional motion detection by averaging both directions (equal weights of 0.5 each). In Fig. 3 (c), the output activations for the two sequences show that the model distinguishes them as the first output is higher for the first sequence, and vice versa.

**Limitations of existing explanation methods.** In Fig. 3 (d), we visualize established XAI methods: attribution maps (Integrated Gradients; Sundararajan et al., 2017) for the target class, and feature visualization for the highest activating patch. For the second output – encoding the *presence* of both directions – the XAI methods faithfully highlight the important image parts to understand the function. For the first output – encoding the *presence* of one direction and the *absence* of the other direction – both XAI methods only highlight the left-to-right movement, *i.e.* the positive potential, and neglect the encoded absence (right-to-left movement), thus not giving a complete explanation.

**Explaining absent features.** In Fig. 3 (e), we visualize our modifications for attribution methods and feature visualization. Computing the *non-target* attribution for the first output with respect to the bi-directional sequence reveals that right-to-left motions have a negative effect on the output (red attribution). Similarly, the *least* activating patch for the first output is a right-to-left movement, faithfully explaining that the first output encodes the *absence* of this direction. For the second output, the attribution for the left-to-right sequence highlights the movement as expected. For the minimally activating patch, we have an activation of zero, and thus, no inhibition is happening and no absence from the dataset is encoded. To conclude, in order to obtain a complete explanation, established and our modified XAI methods have to be used *in combination*, even for this simple model.

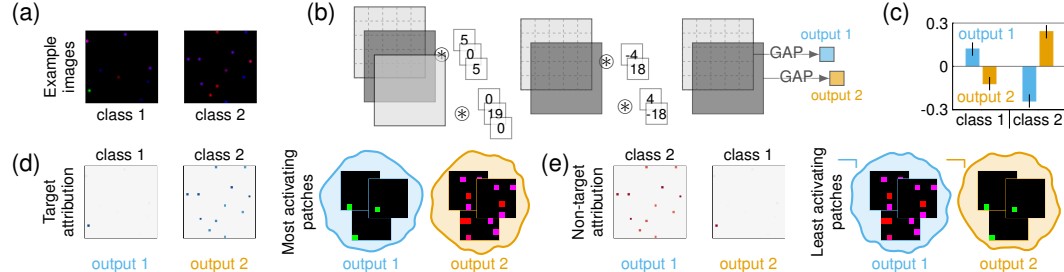

Figure 4: **Toy experiment.** *(a)* Example images from class 1 (green pixel) and class 2 (no green pixel) – zoom in for better visibility. *(b)* Architecture of the used toy model with the trained weights. *(c)* Average logit output for the two output nodes for images from class 1 and 2. Confidence intervals represent two times the standard deviation. *(d)* Integrated Gradients (Sundararajan et al., 2017) target attributions for the above example images, and maximally activating patches for the two output nodes. *(e)* *Non-target* attributions for the two respective examples – note how the attributions switch from positive (blue) to negative (red) – and *minimally* activating patches for the two output nodes.

## 5.2    EXPLAINING ENCODED ABSENCES IN A TRAINED TOY MODEL

We continue with a toy example in which we train a model to classify images based on whether they contain a green pixel (class 1) or not (class 2); see Fig. 4 (a). To ensure that only the presence, respectively, the absence of a green pixel contains class information, the number of non-green pixels is chosen randomly between 8 and 12 for both classes. We use a simple two-layer convolutional neural network with $1 \times 1$ kernels and ReLU activations, followed by global average pooling (GAP). Two scalar outputs indicate whether class 1 or class 2 is predicted – the full model is visualized in Fig. 4 (b); training details are provided in Appendix B.2. The average activation of the two output nodes for images from class 1 (green pixel) and class 2 (no green pixel) in Fig. 4 (c) shows that the second output node has a positive activation if no green pixel is present and a negative activation if a green pixel is present. Thus, according to Definition 2.1, the node encodes the absence of a green pixel. This is further confirmed when analyzing the learned weights in Fig. 4 (b): The first layer learns two features that react to the red/blue channels and the green channel, respectively. The second output node has a positive connection to the channel reacting to red/blue (serving as positive potential) and a negative connection to the channel reacting to green. This exactly reproduces the strategy outlined in Section 2.2, linking our theoretical findings to empirical results and demonstrating that *even a simple DNN is capable of learning to encode the absence of a concept.*

**Limitations of existing explanation methods.**    Now that we know that the first/second output node encodes the *presence/absence* of a green pixel, we again visualize established XAI methods (Integrated Gradients (Sundararajan et al., 2017) and feature visualization) in Fig. 4 (d). Confirming the results from Section 5.1, the first output – encoding the *presence* of a green pixel – can be explained by existing XAI methods. However, for the second output – encoding the *absence* of a green pixel – only the non-green pixels providing the positive potential are highlighted. Thus, one would not be able to link the green pixel to the second output, despite their causal relationship.

**Explaining absent features.**    We conclude our experiment by applying our *non-target* attribution and feature visualization through *minimization* in Fig. 4 (e). For the second output node, now the green pixel is highlighted, faithfully explaining that the node encodes its absence. Interestingly, for the first output node, we observe inhibitory signals from the non-green pixels, indicating that the node has learned to encode the absence of red pixels although they do not contain class-discriminative information – this is further confirmed by looking at the weights of the second layer, which are just mirrored between the two nodes (with different suppression strengths). To summarize, in this and the previous experiment, we have demonstrated that *(i)* DNNs can encode the absence of a concept, *(ii)* established explanation methods struggle to faithfully communicate such encodings, and *(iii)* our proposed modifications are capable of explaining concepts whose absence is encoded.

## 5.3 Explaining Encoded Absences in Image-Classification Models

We now turn to a more realistic setting using ImageNet-1k (Russakovsky et al., 2015) models and test for encoded absences. According to Definition 2.1, a channel encodes the absence of a concept if the presence of the corresponding input patterns decreases its activation. To quantify this effect, we measure the drop in a channel's activation when inserting patches containing such patterns. Specifically, we compare patches obtained via our feature visualization through minimization (least activating) with those derived from the only post-hoc method class we are aware of that has a related motivation, encoded logical NOTs (Mu & Andreas, 2020). To ensure that observed activation drops are not merely due to out-of-distribution artifacts, we additionally evaluate the insertion of random patches that exhibit comparable boundary discontinuities as the other methods. For each channel in the final convolutional layer, we compute the average activation over the 100 most activating images and evaluate the mean drop in activation after inserting $48 \times 48$ patches, either random, encoded logical NOT, or least activating, into a random corner of each image; see Fig. 5 (we report results for alternative hyperparameters in Appendix B.3). Since Mu & Andreas (2020) do not identify a logical NOT for every channel, we evaluate their method only on the subset of channels for which such a concept is found. For a fair comparison, for each of these channels, we again report the mean activations after inserting a random patch as well as after inserting a patch containing the identified logical-NOT concept (random patch → logical-NOT patch in Table 1).

Table 1: **Quantitative evaluation of encoded absences.** We measure the activation of the 100 highest-activating images when inserting none, random, encoded logical NOTs, or least activating $48 \times 48$ patches in a random corner.

| Method | VGG19 | ResNet-50 |
|---|---|---|
| None | 2.98 | 0.18 |
| + Random | 2.68 | 0.16 |
| + Logical NOT | 2.53→2.51 | 0.15→0.15 |
| + Least act. (ours) | 0.94 | 0.03 |

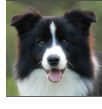 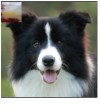 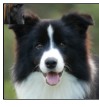

None      +Random      +Least act.

Figure 5: **Example images for Table 1.** Zoom in to see inserted patches.

Table 1 shows that random patches have little effect on channel activation, despite rendering the input slightly out-of-distribution, whereas the least activating patches strongly suppress it, demonstrating their inhibitory role and the encoded absence – an effect that almost *all* channels exhibit (see Appendix B.3). This suggests that encoded absences are indeed utilized in ImageNet models. Logical NOTs identified by Mu & Andreas (2020) show no significant inhibitory effect beyond random patch insertions, consistent with our discussion in Section 3. This empirically supports our claim that their method does not capture encoded absences in the sense established in this work.

**How absences are used.** We now seek to better understand *how* these inhibitory signals are used. While a full mechanistic understanding remains an open challenge and is beyond the scope of this paper, we provide an initial glimpse into the role of absences. To this end, for each class, we find channels in the penultimate layer (other layers could be used too) of a ResNet-50 (He et al., 2016) that are particularly important, similar to Hesse et al. (2025) (see Appendix B.3). Next, for each identified channel, we visualize: *(i)* the maximally activating patches from the respective classes the channel is important for (to show the encoded presences, *i.e.*, the positive potential), and *(ii)* the minimally activating patches (to show the encoded absences). Three illustrative cases are shown in Fig. 6, where channels contribute positively to the class prediction while simultaneously encoding the absence of concepts from closely related classes. To further validate that these minimally activating patches carry meaningful semantics from the model's perspective, we classify each patch using the same ResNet-50 under inspection. In all three groups, at least one minimally activating patch is assigned to a semantically related class: for channel 2026, a patch is classified as "eft" (amphibian), for channel 1470 as "German shepherd," and for channel 494 as "squirrel monkey." These classifications indicate that the patches indeed contain concepts the model associates with related classes, supporting our interpretation that the channel encodes the *absence* of these concepts. While encoding the presence and absence of related concepts that never co-occur might seem redundant (*e.g.*, Border Collie vs. Leonberger snouts; *cf.* Fig. 6, middle), this redundancy can enhance robustness: a Border Collie with a partially occluded snout is more confidently recognized if no Leonberger snouts are detected. Crucially, this phenomenon goes beyond simple redundancy: absences seem to be especially useful for fine-grained classification (*e.g.*, Border Collie vs. Leonberger), where detecting the absence of features from similar classes provides a strong discriminative signal. Although this analysis is qualitative, observing this pattern in three distinct channels is unlikely due to chance alone, given the large number of 1000 ImageNet classes and the correspondingly large number of potential class pairs.

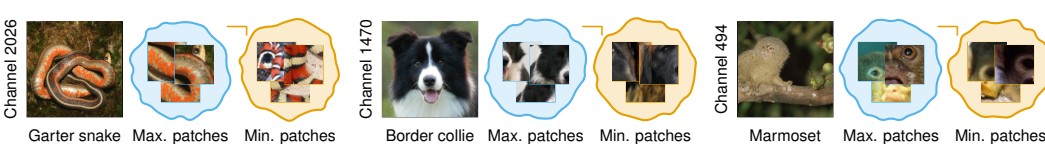

Figure 6: **Encoded presences (positive potential) and absences for three channels that have been found to be important for the corresponding class.** We identify channels that are important for specific classes and visualize the positive potential from this class by showing the most activating patches. Further, we show the encoded absences by showing the least activating patches. In particular, for fine-grained classification, encoded absences of patterns from related species seem to be used.

### 5.4 DEBIASING MODELS BASED ON ENCODED ABSENCES

DNNs are prone to learning spurious correlations in the training data. For instance, in the ISIC dataset (Rotemberg et al., 2021) of skin lesion images, benign samples often co-occur with colorful patches (*cf.* Fig. 7 (a)). Consequently, models trained on this dataset may rely on the presence of colorful patches to classify samples as benign, resulting in biased predictions (Rieger et al., 2020).

We replicate this bias synthetically, allowing for more precise control. Specifically, we generate a training dataset in which all benign samples contain a colorful patch, while malignant ones do not. We train three $\mathcal{X}$-ResNet-50 (Hesse et al., 2021) – models designed for training with attribution priors – with different priors on this biased data and evaluate them on validation sets with varying bias configurations in Table 2. Without any prior/debiasing, the model overfits to the colorful patches and fails when no such patch is available or its association is inverted. Attribution maps (*cf.* Fig. 7 (b) – no debiasing) confirm that the model is focusing on the colorful patch.

To debias such a model, attribution priors (Ross et al., 2017; Rieger et al., 2020) have been proposed. Here, usually, the target attribution for each sample containing a spurious correlation is computed and constrained to be as low as possible in the area of the spurious correlation. When training with such an attribution prior (presence debiasing), the model performs well on validation data without bias, suggesting successful debiasing. However, when the bias is inverted (*i.e.*, benign samples lack colorful patches and malignant ones contain them), the accuracy drops significantly – particularly due to frequent misclassification of malignant samples. As argued in this work, the model may have learned to ignore the presence of colorful patches for benign predictions, yet still relies on their absence to predict malignancy – something not addressed by the attribution prior. This is further supported by the attribution maps in Fig. 7 (b) – presence debiasing: the non-target (malignant) attribution for a benign sample with a colorful patch highlights the patch with negative attribution, indicating that it acts as an inhibitory signal for predicting malignancy.

We, therefore, propose *presence+absence debiasing*: extending the attribution prior to also include our proposed non-target attribution. This effectively suppresses patch attribution for the malignant output on benign samples and prevents the model from using either the presence or the absence of the colorful patch as a shortcut. As a result, we achieve a higher accuracy on the unbiased and inverted-bias validation sets, with attribution maps showing reduced reliance on the patch across both classes (*cf.* Fig. 7 (b) – presence+absence debiasing). Intriguingly, training and evaluating a model on unbiased data – which serves as an upper bound – achieves the same average accuracy as our proposed debiasing, indicating that our strategy successfully removes the bias.[2]

## 6 DISCUSSION

**Limitations and opportunities.** Our non-target attribution approach requires analyzing more attribution maps than standard target attributions, which may limit scalability. For example, in the debiasing experiment in Section 5.4, it required roughly twice the computational effort, and the overhead grows further with the number of classes. To improve scalability and avoid running non-

---

[2]Interestingly, Ross et al. (2017) found that computing attributions with respect to multiple classes slightly improved the stability of their attribution prior, which they attributed to discontinuities near decision boundaries. While this is conceptually similar to our proposed presence+absence debiasing, they did not provide the theoretical insight or additional empirical analyses that we offer here.

Table 2: **Validation results for the ISIC dataset with varying biases.** We report the accuracy (average over 5 runs) for the validation split of the ISIC dataset with different biases. In the "train bias" setup, the train bias is replicated with all the benign samples containing colorful patches, while in the "inverse bias" setup, the malignant samples contain colorful patches, as indicated by $*$. A model with no debiasing learns the dataset bias and fails to classify samples when the bias is *not* present. A model with presence debiasing (existing attribution priors) can reduce this bias; however, it still fails to classify malignant samples when inserting colorful patches, indicating that it is biased based on the *absence* of colorful patches. Our proposed presence+absence debiasing results in the highest average accuracy for both setups without the training bias, and is similarly performant as a model trained without bias, suggesting that the model is largely debiased. "Attr." shows the relative attribution within the colorful patches, confirming qualitative results from Fig. 7.

| Bias | Model | Validation split (train bias) | | | | Validation split (inverse bias) | | | | Validation split (no bias) | | |
|---|---|---|---|---|---|---|---|---|---|---|---|---|
| | | Benign$^*$ | Malignant | Avg. | Attr. | Benign | Malignant$^*$ | Avg. | Attr. | Benign | Malignant | Avg. |
| None | $\mathcal{X}$-ResNet-50 | – | – | – | – | – | – | – | – | 0.84 | 0.77 | **0.81** |
| Benign$^*$ | No debiasing | 1.00 | 0.99 | **0.99** | 0.40 | 0.04 | 0.00 | 0.02 | 0.47 | 0.04 | 0.99 | 0.51 |
| | Presence debiasing | 0.96 | 0.88 | 0.92 | 0.08 | 0.66 | 0.17 | 0.41 | 0.13 | 0.66 | 0.88 | 0.77 |
| | Presence+absence debiasing (ours) | 0.91 | 0.88 | 0.89 | **0.07** | 0.74 | 0.43 | **0.59** | **0.08** | 0.74 | 0.88 | **0.81** |

(a)  (b)

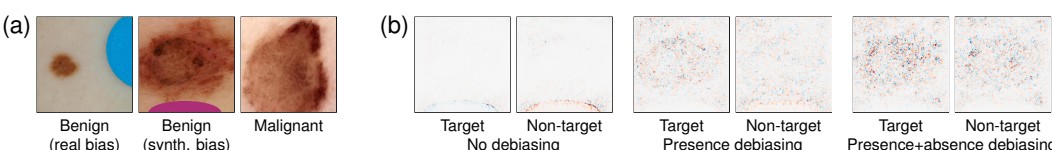

Benign (real bias)  Benign (synth. bias)  Malignant  Target Non-target No debiasing  Target Non-target Presence debiasing  Target Non-target Presence+absence debiasing

Figure 7: **Images and attributions for our biased ISIC dataset.** *(a)* We replicate the ISIC bias (real bias) that co-occurs with the benign samples with a synthetic bias. *(b)* Attributions of different (de)biased models for the benign sample with a synthetic bias. The target attribution is computed for the benign output logit and the non-target attribution for the malignant output. Only including absences in the debiasing prevents the model from relying on patch absence to predict malignancy.

target attributions across all classes, one can first use our feature visualization through minimization to obtain an initial set of encoded absences and then apply non-target attribution only to samples containing these concepts; automatic concept extractors (*e.g.*, Rao et al., 2024) could further reduce redundancy by curating a smaller, diverse set of patches. Further, our experiments deliberately focus on image classification to isolate encoded absences; exploring more complex models or tasks is an important direction for future work. Our findings could be applied wherever inhibitory evidence plays a functional role. For example, in text classification or sentiment analysis, our approach could identify which phrases suppress a particular label; in large-language models, non-target attributions could reveal inhibitory relationships between concepts that influence next-token predictions; and in image generation models, minimization-based feature visualization could help diagnose which visual concepts suppress certain generated attributes, enabling more precise control over model outputs. Finally, we simplify by assuming that concepts are axis-aligned with individual neurons, which is not necessarily true. In Appendix A.2 we outline how to account for this.

**Conclusion.** We show that even concepts not present in the input can affect a neural network's output – a critical but largely overlooked aspect in mainstream XAI. Most established XAI methods struggle to reveal such signals when applied in their standard form, prompting us to propose two simple adjustments to attribution and feature visualization techniques. We empirically show that our non-target attributions and feature visualization through minimization reveal encoded absences and, when combined with existing methods, provide a more complete understanding of DNN behavior. Applying these tools to ImageNet models, we find evidence that encoded absences improve classification, particularly for fine-grained distinctions. Yet, the relevance of encoded absence goes far beyond a mechanistic understanding: we show that biases can stem not only from feature presence but also from their absence, and that effective debiasing must account for this, *e.g.*, through attribution priors on our proposed non-target attributions. While we only take initial steps in this exciting direction, our findings suggest that encoded absences are not only common but as rich and informative as encoded presences. We hope this work opens the door to a broader rethinking of what constitutes an explanation in mainstream XAI.

## REPRODUCIBILITY STATEMENT

We have taken careful steps to make our work fully reproducible. Most importantly, we include the code to reproduce our experiments in the supplement. Additionally, in Appendix B, we provide additional experimental details such as the used framework, GPUs, and hyperparameters for each experiment. For our theoretical results, we clearly specify all assumptions in Section 2.2.

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

# A THEORETICAL ELABORATIONS

Due to space constraints, the main text focuses on presenting our core theoretical insights. Here, we provide additional elaborations to complement the main paper.

## A.1 FEATURE VISUALIZATION AND COUNTERFACTUAL EXPLANATIONS IN THE CAUSAL FRAMEWORK

In Section 2.1 of the main text, we view a DNN $f$ as a structural causal model $\mathfrak{C}$ and argue that the goal of XAI is to find a simplified causal model $\mathfrak{C}'$ that preserves task-relevant causal relationships while improving human interpretability (Hesse et al., 2023; Carloni et al., 2025). Here, we provide a simplified causal model $\mathfrak{C}'$ for two common XAI methods, feature visualization and counterfactual explanations.

For feature visualization (Olah et al., 2017), $\mathfrak{C}'$ reduces the model to a single causal path $z_j := f_j^{(1 \to l)}(x)$ from the input $x$ to a chosen internal neuron $z_j$ in layer $l$, and seeks the input $x = \tilde{x}$ that maximizes the positive activation of the intervention $do(x := \tilde{x})$ on $z_j$.

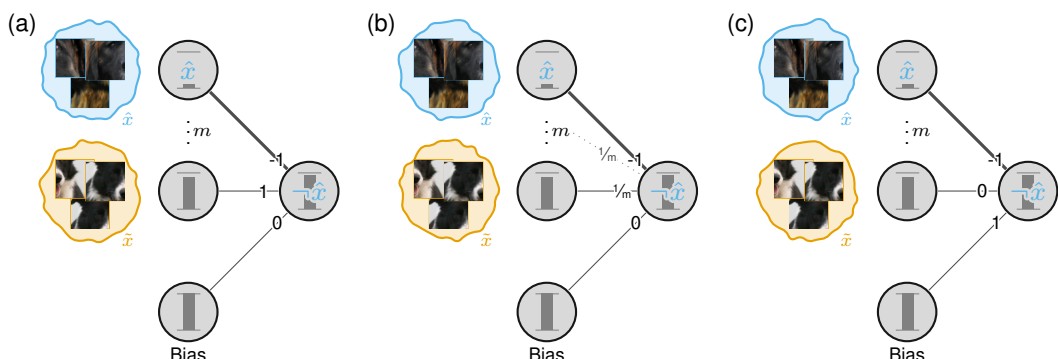

Figure 8: **Three mechanistic processes to encode the absence of a feature.** A neuron encoding the *absence* of latent feature $\hat{x}$ (*i.e.*, $\neg\hat{x}$) can be implemented by having a negative connection to a neuron encoding $\hat{x}$ and a positive potential through *(a)* another activating concept $\tilde{x}$, *(b)* some form of averaging, or *(c)* the bias.

For a counterfactual explanation such as those of Goyal et al. (2019), $\mathfrak{C}'$ simplifies the model to capture the causal relationships necessary to identify the ("minimal") intervention $do(x := \overline{x})$ that changes the prediction from $y = f(x)$ to a desired counterfactual outcome $y' = f(\overline{x})$. This allows us to understand how an input would need to change to result in another prediction or to obtain the class-discriminative features in a sample.

## A.2 ALTERNATIVE IMPLEMENTATIONS FOR ENCODED ABSENCES

In Section 2.2 of the main paper, we outline a specific algorithm for encoding absences – inhibitory activation by the concept whose absence is encoded combined with a positive potential through another concept – that proved particularly relevant in our experimental setting. However, numerous alternatives could be considered, and we outline a few additional examples here.

The positive potential can be implemented in different ways as illustrated in Fig. 8. The positive potential can not only be supplied via *(a)* using the activation of another latent feature $\tilde{x}$ in $l - 1$, but also by *(b)* via a learned averaging over the previous layer (Hesse et al., 2021), or *(c)* via the bias term.

When relaxing the assumptions of unnormalized activations and ReLU activation functions, there are additional strategies to encode the absence of a latent feature $\hat{x}$. For example, instead of having a negative connection from a neuron in layer $l - 1$ encoding the presence of $\hat{x}$ to the neuron $z_j$ in layer $l$ encoding the absence of $\hat{x}$, there could be positive connections to all other neurons but $z_j$. After normalization, the presence of $\hat{x}$ leads to the inhibition of $z_j$, thereby satisfying the condition outlined in Definition 2.1. A neuron that is followed by a symmetric/unbounded activation function, such as Tanh or leaky ReLU, could encode the presence of a feature $\hat{x}$ in the positive direction and its absence in the opposite negative direction, requiring no positive potential. Interestingly, the model could even learn to encode the presence of a feature $\hat{x}$ in the *negative* direction and its absence in the opposite *positive* direction. We leave the identification of such cases to future work. However, once identified, feature visualization by maximization and our proposed feature visualization by minimization must be interpreted inversely to yield the intended explanations.

So far, for simplicity, we have assumed that latent features are axis-aligned with individual neurons. In practice, however, latent features may lie along arbitrary directions in feature space (Elhage et al., 2022; O'Mahony et al., 2023), giving rise to *polysemantic* neurons. Fortunately, our proposed arguments and methods naturally extend to this case by simply substituting "neurons" with "combinations of neurons," respectively, "feature space directions." For example, Definition 2.1 simply becomes:

**Definition A.1 (Encoded Absence For Feature Space Direction)** *If there exists an input pattern/latent feature $\hat{x}$ whose presence causes the activation of a feature space direction $z$ to decrease, we say that the feature space direction $z$ encodes the* absence *of said feature $\hat{x}$.*

Similarly, in our proposed feature visualization through minimization, we could find input patterns that inhibit a specific feature space direction instead of a specific neuron.

The validity of our conclusions is not affected by polysemanticity. Polysemanticity would simply increase the complexity of what a channel encodes. Instead of representing the presence of concepts from one class and the absence of concepts from related classes, as shown in Section 5.3, a polysemantic channel could additionally encode the presence or absence of other, (un)related concepts. This would enrich the interpretation but does not undermine the conclusions we draw.

Please note that finding such meaningful feature space directions is an active area of research (Kim et al., 2018; Fel et al., 2023; O'Mahony et al., 2023) and not the scope of this paper.

## B EXPERIMENTAL DETAILS

In this section, we provide detailed information to facilitate the reproduction of our experiments described in Section 5. All code and trained models will be released under the Apache 2.0 license upon acceptance of the paper. For convenience, the code to reproduce the main results is also included in the supplementary material. All experiments have been run on a single Nvidia A100-SXM4 (80GB) or Nvidia RTX A6000 (48GB) GPU and require only several hours ($\leq 10$) to complete. All code is implemented in PyTorch (Paszke et al., 2017) (3-Clause BSD license). To compute Integrated Gradients (Sundararajan et al., 2017) attributions (zero baseline) in Sections 5.1 to 5.3, we use Captum (Kokhlikyan et al., 2020) (3-Clause BSD license). Please refer to the main paper for an overview of each experiment and additional details.

### B.1 EXPLAINING ENCODED ABSENCES IN A HASSENSTEIN-REICHARDT DETECTOR

As illustrated in Fig. 3 (b), we use a two-layer convolutional neural network with ReLU activation functions for the experiment introduced in Section 5.1. Each layer consists of two channels, with kernel sizes $(C = 2) \times (H = 1) \times (W = 2)$ and $(C = 2) \times (H = 1) \times (W = 1)$, respectively (no bias is used). Since we manually set the weights for the model (see Fig. 3 (b) for exact weights), no training procedure is needed.

The non-target attribution is computed through $\mathcal{A}(x, t', f)$ for both visualized input samples $(x^{(1)}, t^{(1)})$ and $(x^{(2)}, t^{(2)})$, where $t'$ is the complementary class of $t$ in the binary classification setting.

### B.2 EXPLAINING ENCODED ABSENCES IN A TRAINED TOY MODEL

For our toy experiment in Section 5.2, we generate a synthetic training dataset of $20\,000$ images of size $32 \times 32$ containing 8–12 non-green pixels, half of which contain one additional green pixel. Non-green pixels are generated by randomly assigning values of 0, 0.5, or 1 to the red and blue channels, respectively, excluding pure black (*i.e.*, both channels set to zero). The testing dataset contains 1000 images generated in the same fashion. As illustrated in Fig. 4 (b), we use a two-layer convolutional neural network with ReLU activation functions. Each layer consists of two channels, with kernel sizes $(C = 3) \times (H = 1) \times (W = 1)$ and $(C = 2) \times (H = 1) \times (W = 1)$, respectively (no bias is used). We train the model with a binary cross-entropy loss, using an Adam optimizer (Kingma & Ba, 2015) with a learning rate of 0.01 and weight decay of 0.0001; we train for 15 epochs with a batch size of 256. Since the model does not always converge reliably (probably due to its simplicity), we perform five independent training runs and report results based on the best-performing model.

The non-target attribution is computed through $\mathcal{A}(x, t', f)$ for both visualized input samples $(x^{(1)}, t^{(1)})$ and $(x^{(2)}, t^{(2)})$, where $t'$ is the complementary class of $t$ in the binary classification setting.

### B.3 EXPLAINING ENCODED ABSENCES IN IMAGE CLASSIFICATION MODELS

**Quantitative.** For our quantitative analysis of inhibitory signals in ImageNet-trained models, we use the ImageNet-1k validation split (Russakovsky et al., 2015) and PyTorch (Paszke et al., 2017) torchvision models (VGG19 (Simonyan & Zisserman, 2015), ResNet-50 (He et al., 2016)). For each

Table 3: **Quantitative evaluation of encoded absences.** We report the results from Table 1 alongside their standard deviations (indicated by "$\pm$") and under different hyperparameters (patch size and number of images). Please refer to Table 1 for a detailed description.

| Model | Patch size | Nr. images | None | +Random | +Logical NOT | +Least act. |
|---|---|---|---|---|---|---|
| VGG19 (Simonyan & Zisserman, 2015) | 32 | 100 | $2.98 \pm 1.08$ | $2.84 \pm 1.09$ | $2.70 \pm 1.04 \rightarrow 2.68 \pm 1.04$ | $2.14 \pm 1.13$ |
| VGG19 | 48 | 100 | $2.98 \pm 1.08$ | $2.68 \pm 1.10$ | $2.53 \pm 1.06 \rightarrow 2.51 \pm 1.06$ | $0.94 \pm 1.16$ |
| VGG19 | 64 | 100 | $2.98 \pm 1.08$ | $2.41 \pm 1.12$ | $2.25 \pm 1.08 \rightarrow 2.21 \pm 1.08$ | $-0.38 \pm 1.18$ |
| VGG19 | 48 | 50 | $3.72 \pm 1.08$ | $3.39 \pm 1.10$ | $3.21 \pm 1.06 \rightarrow 3.19 \pm 1.06$ | $1.66 \pm 1.16$ |
| VGG19 | 48 | 200 | $2.25 \pm 1.07$ | $1.98 \pm 1.10$ | $1.85 \pm 1.04 \rightarrow 1.84 \pm 1.05$ | $0.25 \pm 1.14$ |
| ResNet-50 (He et al., 2016) | 32 | 100 | $0.18 \pm 0.06$ | $0.17 \pm 0.06$ | $0.16 \pm 0.06 \rightarrow 0.16 \pm 0.06$ | $0.12 \pm 0.07$ |
| ResNet-50 | 48 | 100 | $0.18 \pm 0.06$ | $0.16 \pm 0.06$ | $0.15 \pm 0.06 \rightarrow 0.15 \pm 0.06$ | $0.03 \pm 0.11$ |
| ResNet-50 | 64 | 100 | $0.18 \pm 0.06$ | $0.14 \pm 0.07$ | $0.13 \pm 0.07 \rightarrow 0.14 \pm 0.07$ | $-0.09 \pm 0.11$ |
| ResNet-50 | 48 | 50 | $0.21 \pm 0.05$ | $0.19 \pm 0.06$ | $0.15 \pm 0.06 \rightarrow 0.15 \pm 0.06$ | $0.06 \pm 0.11$ |
| ResNet-50 | 48 | 200 | $0.14 \pm 0.06$ | $0.12 \pm 0.06$ | $0.11 \pm 0.06 \rightarrow 0.11 \pm 0.06$ | $-0.01 \pm 0.11$ |

channel in the last convolutional layer, we identify the 100 images that most strongly activate the respective channel after global average pooling (GAP). To assess the effect of interventions, we modify each of these 100 images by inserting either a random $48 \times 48$ patch, a patch containing the concept of the logical NOT (Mu & Andreas, 2020), or one of the eight least activating $48 \times 48$ patches into a randomly selected corner of the image. To find the least activating patches, we use a sliding-window approach with a stride of 16. For identifying logical NOTs from Mu & Andreas (2020), we use the default hyperparameters with the only exception of reducing the beam search limit to 50, which was recommended by the authors for getting good explanations in a reasonable time. We compute the average channel activation (after GAP) across all modified images and all channels. In Table 3, we report the mean activation values from Table 1 alongside the corresponding standard deviations. Since Mu & Andreas (2020) do not identify a logical NOT for every channel, we evaluate their method only on the subset of channels for which such a concept is found. For a fair comparison, for each of these channels, we again report the mean activations after inserting a random patch as well as after inserting a patch containing the identified logical-NOT concept (random patch $\rightarrow$ logical-NOT patch). We additionally test different hyperparameter configurations and observe the same pattern: in both models, there exist patches that inhibit the activation of specific channels, indicating that the models utilize encoded absences (*cf.* Section 5.3). To further assess the statistical significance of our findings, we perform a $t$-test (SciPy's scipy.stats.ttest_ind) comparing activations for images with randomly inserted patches to those with the lowest-activating patches inserted. For both models, the resulting $p$-values are close to zero ($\sim 10^{-91}$ for VGG19 and $\sim 10^{-125}$ for ResNet-50), indicating statistical significance.

To better understand *how many* channels encode absences, we further measure the fraction of channels in the final convolutional layer of the analyzed models that are statistically significantly affected by an inhibitory effect (*i.e.*, where the activation of a channel differs between images with the least activating patch inserted and those with random patches). Remarkably, this holds for $512/512$ channels in VGG-19 and $2036/2048$ channels in ResNet-50. Thus, almost *all* channels encode absences, indicating that this phenomenon is a systematic property of image classification models and warrants further investigation.

To further validate our evaluation protocol, we also tested the opposite case by inserting *maximally* activating patches. As expected, the activations increase (VGG19: $2.98 \rightarrow 3.88$; ResNet-50: $0.18 \rightarrow 0.25$).

**Qualitative.** To find the qualitative examples from Fig. 6, we start by computing Integrated Gradients (Sundararajan et al., 2017) attributions for each output logit with respect to the last convolutional layer of the above ResNet-50 (He et al., 2016) trained on ImageNet, using all validation samples of the corresponding class. Other layers, besides the penultimate one, could also have been used – later layers are likely to capture more high-level semantic features and may therefore be better suited for our analysis. We discard negative attributions because, for now, we focus only on channels that positively contribute to class prediction – *i.e.*, channels whose presence is important for predicting the class. We then average the attributions across samples. Channels are considered *important* for a specific class if their relative attribution (*i.e.*, attribution divided by total class attribution) is at least 0.05. For each channel that is important for a specific class, we obtain the most activating patches for images from that class to visualize the encoded *presence*, respectively, the positive potential. Now that we know that the channel is important for predicting the class of interest and which *presences* cause it to activate, we aim to find which absences it encodes. To this end, the least activating patches for that channel are extracted from the entire validation split. For both the most and least activating patches,

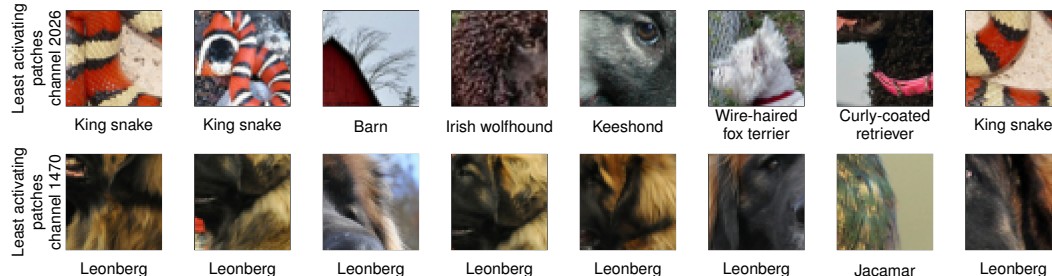

Figure 9: **The eight least activating patches for channels 2026 and 1470.** For the most and least activating patches in Section 5.3, we obtain eight candidate patches and manually select a monosemantic subset of three patches for more interpretable visualizations. Inspecting all eight patches for channels 2026 and 1470 in ResNet-50 reveals that these channels encode the absence of *multiple* concepts, consistent with prior work on polysemantic neurons Elhage et al. (2022). The corresponding labels indicate the class of each patch.

we extract the eight most/least activating candidate patches. We then manually select a monosemantic subset of three patches for more interpretable visualizations. While this manual selection does not affect the validity of our conclusions, it may convey a more monosemantic impression than is accurate – additional concepts may be present among the full set of eight patches (see Fig. 9) as was discussed as a limitation in the main paper.

### B.4 DEBIASING MODELS BASED ON ENCODED ABSENCES

For our debiasing experiment in Section 5.4, we use the ISIC 2020 dataset (Rotemberg et al., 2021, CC-BY-NC license) of skin lesion images. Since the dataset is heavily imbalanced, with more benign than malignant samples, we randomly subsample the splits to create balanced sets, resulting in a training dataset of 1168 samples and an evaluation dataset of 524 samples. To increase the diversity of the samples, we apply random flipping and color jittering (brightness=0.2, contrast=0.2, saturation=0.2). The used $\mathcal{X}$-ResNet-50 model (Hesse et al., 2021) is pre-trained on ImageNet-1k (Russakovsky et al., 2015), with weights obtained from (Hesse et al., 2021, Apache-2.0 license). We finetune each model with a binary cross-entropy loss, using an Adam optimizer (Kingma & Ba, 2015) with a learning rate of 0.0001 and weight decay of 0.0001; we train for 20 epochs with a batch size of 128. The loss of the models with *no debiasing* on the unbiased and biased datasets can be written as

$$\mathcal{L} = \text{BCE}(x, t, f), \tag{1}$$

with BCE denoting the binary cross-entropy loss, $x$ the input sample, $t$ the target label, and $f$ the model. When training with *presence debiasing*, the loss becomes

$$\mathcal{L} = \text{BCE}(x, t, f) + 2\lambda \frac{|\mathcal{A}(x, t, f)\mathcal{P}(x)|}{|\mathcal{P}(x)| + 10^{-5}}, \tag{2}$$

with $\mathcal{A}(x, t, f)$ denoting the input attribution (Integrated Gradients; Sundararajan et al., 2017) and $\mathcal{P}(x)$ the segmentation mask of the colorful patch with 1 indicating its presence and 0 its absence; we dilate the mask by 10 pixels to include the edges. To prevent division by zero for images that contain no colorful patch, we add $10^{-5}$ to the denominator. We weight the attribution prior with a factor of 2 to account for the double attribution prior used in *presence+absence debiasing*, allowing for a fairer comparison. For *presence+absence debiasing*, the loss can be formulated as

$$\mathcal{L} = \text{BCE}(x, t, f) + \lambda \left( \frac{|\mathcal{A}(x, t, f)\mathcal{P}(x)|}{|\mathcal{P}(x)| + 10^{-5}} + \frac{|\mathcal{A}(x, t', f)\mathcal{P}(x)|}{|\mathcal{P}(x)| + 10^{-5}} \right), \tag{3}$$

with $t'$ being the complementary class of $t$ in our binary classification setting. In this experiment, only benign samples contain colorful patches during training, which means that the attribution prior for malignant samples is always zero ($|\mathcal{P}(x)| = 0$). Consequently, in all cases where the attribution prior has an effect, the true label $t$ corresponds to the benign class, and the complementary label $t'$ corresponds to the malignant class. Intuitively, in the *presence+absence debiasing* procedure, we

Table 4: **Validation results for the ISIC dataset with varying biases.** We report the results from Table 2 together with their standard deviations (indicated by "±"). Please refer to Table 2 for a detailed description of the table.

| Bias | Model | Validation split (train bias) | | | | Validation split (inverse bias) | | | | Validation split (no bias) | | |
|---|---|---|---|---|---|---|---|---|---|---|---|---|
| | | Benign* | Malignant | Avg. | Attr. | Benign | Malignant* | Avg. | Attr. | Benign | Malignant | Avg. |
| None | $\mathcal{X}$-ResNet-50 | – | – | – | – | – | – | – | – | 0.84 | 0.77 | **0.81** |
| | | – | – | – | – | – | – | – | – | ± 0.03 | ± 0.07 | ± 0.02 |
| Benign* | No debiasing | 1.00 | 0.99 | **0.99** | 0.40 | 0.04 | 0.00 | 0.02 | 0.47 | 0.04 | 0.99 | 0.51 |
| | | ± 0.00 | ± 0.02 | ± 0.01 | ± 0.02 | ± 0.02 | ± 0.00 | ± 0.01 | ± 0.02 | ± 0.02 | ± 0.02 | ± 0.00 |
| | Presence debiasing | 0.96 | 0.88 | 0.92 | 0.08 | 0.66 | 0.17 | 0.41 | 0.13 | 0.66 | 0.88 | 0.77 |
| | | ± 0.01 | ± 0.10 | ± 0.05 | ± 0.00 | ± 0.08 | ± 0.05 | ± 0.03 | ± 0.05 | ± 0.08 | ± 0.10 | ± 0.01 |
| | Presence+absence debiasing (ours) | 0.91 | 0.88 | 0.89 | **0.07** | 0.74 | 0.43 | **0.59** | **0.08** | 0.74 | 0.88 | **0.81** |
| | | ± 0.06 | ± 0.01 | ± 0.03 | ± 0.01 | ± 0.06 | ± 0.06 | ± 0.03 | ± 0.01 | ± 0.06 | ± 0.01 | ± 0.03 |

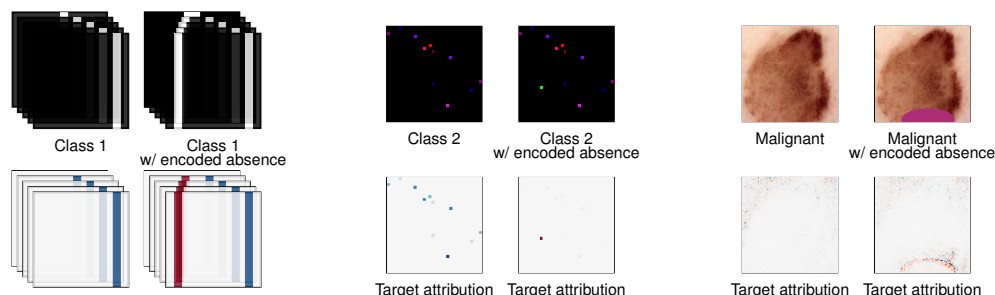

Figure 10: **Controlled concept intervention.** For each experiment relying on non-target attributions, we select a sample from the class that is hypothesized to use an encoded absence (left-side sample of each group). The target attributions for these unmodified samples show no strong inhibitory signals, as indicated by the absence of pronounced red highlights. After inserting the corresponding absence concept into the right-side samples (a right-to-left motion pattern in the Hassenstein–Reichardt detector example; a green pixel in the toy example, bottom-left quadrant; and a colorful patch in the malignant sample from the debiasing experiment) and recomputing the target attribution (see Appendix B.5 for details), we observe markedly stronger inhibitory responses, visible as strong red attributions. This controlled intervention provides clear evidence that the inhibitory patterns identified in our experiments indeed reflect encoded absences.

compute the attribution for the malignant label on benign samples with colorful patches in order to assess the influence of these patches on malignant predictions. Each model is trained for 5 runs, and we determine the prior strength $\lambda \in \{1, 10, 100, 1000, 10000\}$ such that the resulting model performs the best on unbiased data. In Table 4, we expand on the results from Table 2 in terms of their standard deviations.

### B.5 CONTROLLED CONCEPT INTERVENTION

As outlined in Section 4, negative non-target attributions do not in general guarantee the presence of encoded absences, which could in principle affect our experimental conclusions. To verify that this issue does not influence our findings, we conduct a qualitative analysis in Fig. 10, comparing attributions for the same sample before and after introducing the encoded absence.

For each experiment from Sections 5.1, 5.2 and 5.4 that relies on non-target attributions, we take a sample from the class where the model is hypothesized to rely on an encoded absence: class 1 containing a left-to-right movement in the Hassenstein–Reichardt detector, class 2 containing no green pixel in the toy example, and a malignant sample containing no colorful patch in the debiasing experiment (left sample of each experiment in Fig. 10). For each such sample, we compute the *target* attribution (shown below the respective sample in Fig. 10). These attributions exhibit no strong negative regions (no strong red highlights), indicating that no or only minimal inhibitory signals are present in the unmodified inputs where the concept of the encoded absence is absent.

We then insert the concept corresponding to the encoded absence into the same samples (right sample of each experiment in Fig. 10). Specifically, we include a right-to-left movement in the Hassenstein-Reichardt detector sample, a green pixel in the sample from the toy experiment (bottom left quarter; zoom in), and a colorful patch in the malignant sample from the debiasing experiment (bottom). We again compute the target attribution on these samples (shown below the respective sample in Fig. 10). The inserted concepts produce substantially stronger negative attributions (red) than observed in the original samples. This confirms that the negative attributions used in our experimental sections genuinely arise from encoded absences rather than from unrelated effects.

Note that in this controlled setup, we use *target* attributions rather than non-target attributions. This is because the intervention explicitly inserts the concept whose absence is encoded into a sample from the class of interest, turning the input into one that now contains that concept. In such a setting, where the concept is present by construction, target attributions are the appropriate choice. In typical real-world scenarios and in our main experiments, we do not have access to such explicit concept insertions. The class of interest usually does not contain the concept whose absence is encoded, and the goal is precisely to detect how the model responds to that absence. As a result, non-target attributions must be computed on samples from *other* classes that do contain the concept, enabling us to identify the inhibitory relationship.

## C   BROADER IMPACT

Since our method is largely independent of any specific downstream application – such as image generation or facial recognition – we do not anticipate *direct* negative societal impact in such domains. However, our approach contributes to a deeper understanding of neural networks, which could carry *indirect* risks. For instance, improved insights into model behavior might enable the extraction of sensitive information from training data. Similarly, an enhanced mechanistic understanding could potentially be misused to manipulate models into producing targeted outputs, akin to adversarial attacks. Moreover, while we demonstrate how our method can be applied to debias models, it is conceivable that the same techniques could be reversed to intentionally introduce bias.

While these risks are important to acknowledge, a more comprehensive understanding of model behavior also enables substantial positive societal impacts. These include the ability to debias models by reducing their reliance on sensitive or critical features, foster trust in machine learning systems, and identify and mitigate model vulnerabilities or limitations.

If our algorithm does not function as intended – for example, in models with symmetric activation functions where it is unclear if the absence of a feature is encoded in positive or negative activations – it could lead to incorrect interpretations and, consequently, incorrect adjustments applied to the model. It is, therefore, crucial to be aware of the theoretical limitations of the method and, in cases of uncertainty, to conduct deeper analyses of the model to ensure a correct understanding of how a given neuron behaves.

## D   THE USE OF LARGE LANGUAGE MODELS

In the preparation of this work, we made use of large language models (LLMs), in particular ChatGPT and Grammarly, to support various stages of the writing and research process. Specifically, these tools were employed to *(i)* improve the clarity and readability of the text by polishing language and style, and *(ii)* assist in literature review by suggesting potentially relevant related work to ensure that no important references were overlooked. All substantive contributions, including the initial idea, the design of the research, analysis, and interpretation of results, have been conducted by the authors.

For full transparency, we note that this declaration itself was written with the assistance of an LLM.

