# OpenReview forum: "What is Missing? Explaining Neurons Activated by Absent Concepts"
_ICLR.cc/2026/Conference — Submitted to ICLR 2026_

### Official Review · Reviewer_iv67 · 2025-10-23

**Soundness:** 3
**Presentation:** 3
**Contribution:** 3
**Rating:** 4
**Confidence:** 3

**Summary:**

The paper argues that the common XAI approach assumes that neurons activate for current concepts and largely ignores neurons whose activation increases when a concept is absent.  The authors claim that they formalize the so-called encoded absence causality,  explain how and why commonly used attribution and feature visualization methods miss the point, and propose simple extensions. e.g., not-target attribution and the so-called feature visualization by minimization to reveal relationships. They include experimentation results on a couple of datasets, including ImageNet.

In this reviewer's opinion, the draft motivates both methods and explicitly positions them as responses to a gap in current XAI evaluation protocols. In the authors' opinion, currently used protocols assess explanations only with respect to the given input, thus failing to evaluate concept absences.

**Strengths:**

1. In this reviewer's opinion, the clear, causal definition of “encoded absence” and framing of absences as first-class explanatory objects, not merely negative saliency, represent a contribution as it reframes minimization and multi-class saliency as tools for surfacing inhibitory evidence rather than merely as technical variants.

2. I should also state that I found the "mechanistic demonstrations" to be useful rather than cosmetic. For example, in Figure 3, we see that a Hassenstein–Reichardt-style toy is implemented, where the first output unit encodes the absence of right-to-left motion; established target attributions and max-patch visualization fail there, while the proposed non-target attribution and minimization succeed. The green-pixel toy then shows a learned unit with positive weights to red/blue and negative to green, cleanly matching the “encoded absence” construction.

3. The draft self-acknowledges scope limits, such as axis-aligned neuron assumption, CNN focus, scalability of non-target attribution, and deferred LLM/ViT generality. I appreciate the authors' candour and their choice not to oversell the contribution.

**Weaknesses:**

1. In this reviewer's opinion, there is a "mental"  leap. The connection between negative evidence to encoded absence is not fully causal. Non-target negative attributions on out-of-class inputs conflate many forms of anti-evidence. The draft paper sketches a causal definition (see definition 2.1, referenced around the non-target discussion). However,  the experiments rarely isolate just the hypothesized concept while holding all else constant. The authors may want to explore the issue further. Controlled counterfactuals may be useful in that regard. Just one suggestion, although other options may be more useful.

2. The ImageNet “least-activating patch insertion” result risks a distribution shift. Corners and copy-paste borders can depress activations for the wrong reasons.  In my view, copying and pasting “least-activating” patches into corners could introduce artifacts, such as edges and texture breaks. The strong suppression may partly be from low-level off-manifold cues rather than true concept absence encoding. It's not clear in the current draft.

3. Unless I am missing something, the papers note that common protocols only evaluate with respect to pixels present in x. However,  the experiment section does not introduce a general absence-aware metric beyond patch suppression.

**Questions:**

See commentary on weaknesses.

---

> ### Author Response · Authors · 2025-11-25
>
> We thank the reviewer for the thoughtful and constructive feedback, and for highlighting the strengths of our causal framing and mechanistic demonstrations.
>
> **1. Other factors of negative non-target attributions.** We thank the reviewer for highlighting this important nuance. We agree that negative attributions can arise for reasons other than inhibitory evidence (e.g., gradient shattering), and we now explicitly note this limitation in the revised manuscript (L. 234-239). To directly address the reviewer’s concern about causal isolation, we added a controlled interventional experiment in Appendix B.5 following the reviewer’s suggestion (see updated paper).
>
> In this experiment, we isolate the effect of adding a concept of an encoded absence and confirm that these interventions are the main driver for strong negative attribution. This suggests that our observed effects are really due to encoded absences and not other reasons.
>
>
> **2. Distribution shift in the ImageNet least-activating patch insertion.**
> We appreciate this careful observation. To address exactly this concern, our original submission already included a control experiment comparing the least-activating patches to *random* patches with similar edge and texture discontinuities. Specifically:
>
> - We randomly sampled patches that, when pasted into image corners, introduce **comparable edge and texture discontinuities**.
> - We then compared the resulting activation changes to those induced by our *least-activating* patches.
>
> As reported in Appendix B.3, for almost all channels, the least-activating patches yield a **significantly larger** activation drop than random patches of the same shape. This suggests that the observed activation drops are driven by genuine inhibitory effects and not by out-of-distribution effects.
>
> We now explicitly explain this rationale and the role of random-patch baselines in Section 5.3 (L. 390-394) of the revised manuscript.
>
> **3. Missing absence-aware metric.** We agree with the reviewer’s observation. Our discussion in the related-work section was intended only to highlight that many popular XAI evaluation protocols focus solely on evidence *present* in an input, thereby overlooking inhibitory evidence. We did **not** intend to imply that our work introduces a new absence-aware benchmark. To avoid this confusion, we have removed this paragraph from the related-work section and simplified the framing.

---

> > ### Comment · Reviewer_iv67 · 2025-11-26
> > **What is Missing? Explaining Neurons Activated by Absent Concepts**
> >
> > Thanks for the detailed answers and for taking the time to respond. I appreciate the time and effort, but my overall assessment does not change. I will maintain my score.

---

### Official Review · Reviewer_A8Sw · 2025-10-31

**Soundness:** 3
**Presentation:** 4
**Contribution:** 3
**Rating:** 8
**Confidence:** 3

**Summary:**

The authors propose modest modifications to existing interpretability and visualization methods in order to enable explainability from encoded absences. Specifically, they propose non-target attributions (extending from target attributions) and feature visualization through minimization (extending from maximization) via patches.

They provide an illustrative toy example for each extension as well as a more realistic performance comparison (where previous XAI methods can’t capture encoded absences) on a real image dataset. They also show how these enhanced understandings of interpretability which include encoded absences enable improved debiasing of image classification models, with another realistic example.

**Strengths:**

Kudos to the authors for a well-written, compact, and conceptually clean paper that executes on a simple, but good, idea. The paper is easy to follow, which is important for adoption of the methodological extensions in various applications. It is about as thorough in explaining and demonstrating the utility of the proposed extensions as I can expect a 9 page paper to be, from providing carefully designed, illustrative toy examples to showing performance on more realistic image datasets. The debiasing example on real images is compelling. The Appendix helps the work feel complete, particularly the documentation of the experiments.

The figures greatly help the reader understand the core contribution of the paper. Fig 4, Fig 5. - appreciated the notes to “zoom in”.

**Weaknesses:**

Since the (two) updates to existing methods feel somewhat modest, my biggest concern is whether the scope of contribution is large enough for a main paper. However, the simplicity is also appealing – it’s a good idea!

The Intro overpromises application in other fields (you name drop “biological neural networks”) but the paper doesn’t illustrate a general use case beyond images. Furthermore, the Appendix section C Broader Impacts doesn’t address what domains you would imagine your proposed methods to be useful in. I think you can be more honest about the scope illustrated in the paper while also suggesting specific problems and use cases for your methods in other domains.

**Questions:**

Questions

Do you expect your proposed methods to have a use beyond images? Can you make this clearer in the Intro/Discussion/C Broader Impacts by specifically stating fields and problems where the absence of features can be important? Having a recommendation for where to apply these methodological extensions next would be great.



Minor comments not affecting score, suggestions to the authors

In the introduction, so many key words are italicized that it’s hard to know which are most important - would reconsider. Italicizing the same heavily used word (e.g. presence/present) over and over does not add value or clarity. Just once, the first time, is sufficient. The rest of the paper seems to italicize appropriately.

Additionally, italicizing the word “biological” felt misleading since the paper didn’t contain a good biological example other than the older, cited Drosophila anecdote.

299-300 missing an “or”?

---

> ### Author Response · Authors · 2025-11-25
>
> We thank the reviewer for the thoughtful and encouraging feedback, and we address all concerns below.
>
> **1. Is the scope of the contribution large enough for a main paper?**
> We appreciate the reviewer’s observation that the methodological updates alone appear modest. We would like to emphasize that the main contribution of the paper is **conceptual rather than purely algorithmic**. Beyond introducing two extensions of existing methods, our work contributes the four points listed in the reply for Reviewer 1bAo to points 1.2 and 1.3.
>
> To the best of our knowledge, no previous work connects these algorithmic variants to a causal notion of encoded absence, nor demonstrates that this perspective yields practical benefits such as improved debiasing. In line with Reviewer 1bAo’s assessment, we now emphasize this conceptual thread more clearly in the revised Introduction and Sections 2-4.
>
> **Extension to other domains.**
> We thank the reviewer for noting that the biological example in the Introduction may set a wrong expectation. Its purpose was to give intuition. We have removed the italics on *biological* and clarified that the phenomenon is discussed in the context of **artificial neural networks**.
>
> Regarding applications beyond images, in Sec. 6, we now highlight several domains where encoded absences may be particularly useful: other classification domains, large-language models, and image generation models (see L. 520-524 in the revision for more details).
>
> **Minor points.** We thank the reviewer for the recommendations and have included them.

---

> > ### Comment · Reviewer_A8Sw · 2025-11-26
> >
> > Thanks for the clarifications and small changes. I maintain my score.

---

### Official Review · Reviewer_GVJY · 2025-10-31

**Soundness:** 3
**Presentation:** 3
**Contribution:** 3
**Rating:** 6
**Confidence:** 4

**Summary:**

The paper investigates the concept of encoded absences in neural networks within the context of explainable AI (xAI). It introduces adaptations of existing xAI methods to visualize image structures that suppress specific classes or features. In the experiments, the authors demonstrate the approach using two toy examples and evaluate it on an image classification task with VGG-19 and ResNet-50 models. They further illustrate its applicability in a debiasing context.

**Strengths:**

- The paper is well written and presents the concept of **encoded absences** in a logical and structured manner, which helps the reader understand its significance.
- The experimental design is well-thought-out: the authors begin with conceptual and toy examples, then progress to image classification and finally to a debiasing task. This gradual development effectively builds understanding.
- The **debiasing task** is particularly valuable, and the authors propose a **promising** approach to addressing this important problem.

**Weaknesses:**

- Similarity to other xAI methods: While the paper introduces a novel analytical perspective, some of its ideas resemble concepts explored in existing explainable AI methods. For example, the notion of **negative features**—or features contributing against a prediction—has been addressed in approaches such as **Shapley values** [1]. Furthermore, the idea of leveraging the **entire dataset** to obtain a global understanding of both positive and negative influences aligns with the concepts of **prototypes and criticisms** introduced by Kim et al [2]. (2016).  I do see the value in incorporating information from **other classes** to explain the **absence of features** in the target class; however, I feel this idea could potentially be **explored using existing techniques**.

- Experiment's motivation: The purpose of the experiment in **Section 5.3**, particularly the use of a **minimization patch**, is not entirely clear. It seems expected that such a patch would decrease the activationas you specifically chose to include a patch that minimizes activation. Was this measured at the **last layer**? Additionally, it would be interesting to test the opposite case: what happens if you insert a highly activating patch (e.g., replacing the Border Collie’s snout with that of a Leonberger)?

- Experimet’s methodology: The examples in **Figure 6** appear to have been selected a monosemantic subset of patches based on **visual similarity**, which is inherently qualitative. Was there any **systematic human evaluation** to support this categorization? Moreover, it could be insightful to **classify these patches individually** to examine whether they carry recognizable semantic meaning for the network—this could make the conclusions more robust.

- Discussion: Regarding **Figure 6**, were the channels shown important only for this specific class? If so, how does the **polymorphism of channels** (i.e., channels being important for multiple classes) affect the analysis? Clarifying this would strengthen the interpretation.

[1] Lundberg, S. M., & Lee, S.-I. (2017). *A Unified Approach to Interpreting Model Predictions*. Advances in Neural Information Processing Systems (NeurIPS).

[2] Kim, B., Khanna, R., & Koyejo, O. (2016). *Examples are not Enough, Learn to Criticize! Criticism for Interpretability*. Advances in Neural Information Processing Systems (NeurIPS).

**Questions:**

- The discussion on the limitations of existing explanation methods appears throughout the paper. To improve readability and structure, it may be more effective to consolidate these points into the state-of-the-art (SOTA) section instead.
- I include some questions above.

---

> ### Author Response · Authors · 2025-11-25
>
> We thank the reviewer for the thoughtful and encouraging feedback, and we address all concerns below.
>
> **1.1 Relationship to negative attributions.** We thank the reviewer for pointing out the connection to negative attributions such as SHAP [1]. As noted in the initial manuscript (now L. 184-188), negative attributions *are* a form of inhibitory signal, and we have added a reference to [1]. Our point is that most attribution work for images either uses absolute values or focuses on the target sample alone, thereby ignoring negative evidence from **other classes**. This cross-class negative evidence is essential for detecting encoded absences, and to the best of our knowledge, no existing method uses it for this purpose.
>
> **1.2 Relationship to Kim et al., 2016 [2].** We appreciate the reviewer drawing attention to this connection. While [2] does use more than one sample for building an explanation, their method focuses on *prototypes* and *criticisms*, i.e., in the case of their ImageNet experiment, samples from the same class that are poorly captured by the found prototypes. For example, in their Fig. 2, both the prototypes and criticisms are dogs from the *same* class. This form of criticism highlights corner cases within a class but does not identify inhibitory signals originating from the **presence of concepts in other classes**, which is central to our definition of encoded absence. We have added this clarification to Sec. 3, L. 167-170.
>
> **2. Motivation for using the least activating patches in 5.3.** The least activating patch is simply the result obtained by using our proposed feature visualization through minimization (we made this explicit in the revision; e.g., L. 387). While it is expected that such patches reduce activations, this is precisely the behavior that characterizes an **encoded absence** under Definition 2.1: the *presence* of a concept that causally *suppresses* a channel. Thus, this experiment evaluates exactly what our definition requires.
>
> To put this effect into context, we added a baseline using explanations through logical compositions, including NOTs (see our reply to Reviewer 1bAo). These explanations aim to capture logical negation, yet when inserting these logical NOT concepts, they show almost no inhibitory effect beyond random patch insertion. This contrast underscores that our minimization-based visualization is capturing a phenomenon that these methods do not, and demonstrates the utility of our evaluation.
>
> We use the **final convolutional layer** for evaluation (L. 393-394).
>
> At the reviewer’s request, we also tested the opposite case: inserting *maximally* activating patches increases activation, as expected (e.g., VGG19: 2.98 → 3.88; ResNet-50: 0.18 → 0.25). We have included this analysis in App. B3, L. 844-846.
>
> **3. Methodology for Fig. 6.** We did not conduct a human evaluation. However, given the 1000 ImageNet classes and the correspondingly large number of potential class pairs, observing the same pattern – presence of target concepts and absence of related-class concepts – in three independent channels is unlikely due to chance alone, which we now note in the revision (L. 430-432).
> Following the reviewer’s suggestion, we classified the minimally activating patches (L. 419-425) and found that, for each channel, at least one patch is assigned to a semantically related class, indicating that these patches carry meaningful semantics for the model.
>
> **4. Polysemanticity of channels.** The shown channels are indeed important for multiple classes. This does not weaken our conclusion: polysemanticity simply **adds additional encoded relationships**. Instead of encoding only presences and absences for a single class pair, a channel may encode presences/absences for several additional (un)related concepts. We have clarified this in our discussion on polysemanticity in App. A.2, L. 754-757.

---

> > ### Comment · Reviewer_GVJY · 2025-11-26
> >
> > Thanks for the detailed answers — they helped clear up my questions. The added explanations and experiments make the paper stronger and clarify how it differs from related work. There are still a few things that could be explored more in the future, but they don’t change my overall impression. I will increase my score.

---

### Official Review · Reviewer_1bAo · 2025-11-01

**Soundness:** 1
**Presentation:** 2
**Contribution:** 1
**Rating:** 2
**Confidence:** 5

**Summary:**

This paper studies the phenomenon of encoded absences, which occurs when deep neural networks detect or use the absence of a given concept during inference. The paper first highlights some interpretability frameworks and explains why those frameworks do not or cannot capture encoded absence in their explanations. Then, it proposes two extensions for feature attribution and feature visualization aiming to capture this phenomenon. The feature visualization extension consists of an adaptation of the activation maximization method, where minimization is used in place of maximization. For feature attributions, the authors propose a “Non-target” attribution, where the attribution of the original sample is compared against the attribution of a (or multiple) sample of a different class but with respect to the target class. The paper presents applications in image classification and an extension to the debiasing case, where including absence information in the loss function improves model performance.

**Strengths:**

- With some exceptions about missing details (see below), the paper is well written
- The application of encoded absence to debiasing techniques seem a promising and interesting future direction

**Weaknesses:**

- **Novelty, Significance, and Contribution**: Both the concept of encoded absence and the extensions proposed by the authors have already been studied in the literature. For the extensions, as the authors noted, the algorithms are unchanged. Therefore, there is not enough contribution in the paper to support publication. More details are provided below:
   - At the conceptual level, the concept of **encoded absence in activations and neurons has already been extensively studied**. The authors attempt to distinguish their approach from counterfactuals by stating that *“our approach is applied on a neuron level allowing for a mechanistic understanding”* However, encoded absence has already been examined in the context of neuron explanations. For example, previous work (Mu et al [1]) explicitly includes the NOT operator in neuron explanations, and compositional explanations reflect the absence of certain concepts. The class of compositional explanations has been widely explored and extended in the literature. The same is true for the property of *“whose absence causes high activations, or vice versa, whose presence strongly suppresses the activation of a specific internal”* These properties have been demonstrated in several works (e.g., [2, 3]), so sections 2.1 and 2.2 do not represent novelty at the conceptual level. Note that the lack of acknowledge of this area of work induced authors to **several overclaims** over the text about proofs related to this phenomenon.
   - Both the **proposed extensions have also been previously introduced and widely studied in the literature**. The minimization objective for feature visualization is a well-known method, **as even acknowledged by the authors**. The same applies to feature attribution, where similar mechanisms have been used and proposed before. The authors argue that *”while these methods share the same underlying algorithms, their intent and interpretive framing differ fundamentally”*. From the reviewer’s perspective, this is not enough to substantiate a contribution. In fact, the proposed “interpretative framing” is tied to the concept of absence, which is itself not novel, as previously described.
   - To the best of my knowledge, the only novel application appears in the debiasing experiments (but I could be wrong). However, the paper’s narrative revolves around neurons and the concept of absence in general, which is not novel. The debiasing section is not presented as a main contribution and comprises only a small part of the experiments.

Based on this analysis, this is the paper’s main issue, which cannot be resolved without a complete change of narrative. Other concerns are related to the following areas:

- **Missing details and structure**: Several details are missing in the main text (and the appendix does not clarify them). For example, how are non-target attributions computed at the mathematical level? Is it just a visual comparison of the same sample but with attributions to different classes (as seems to be the case from the code), or is there a mathematical process to select specific samples from other classes for the explanations? How are these samples selected? Is it just a visual combination over the full dataset?

- **Overgeneralized statements**: There are several broad statements such as *“standard XAI methods fail to explain encoded absence”* that are vague and inaccurate. The authors discuss a few methods in section 4, highlighting their weaknesses with respect to encoded absence. However, the list is not exhaustive, and from a scientific perspective, listing all the methods that do not address a phenomenon is not informative. It would have been more useful to discuss methods similar to (or approaching) the proposed approaches and the differences among them. The current list omits several relevant works and includes some odd citations. For example, self-explainable DNNs are included but not relevant in later sections or experiments, and some relevant references (e.g., [4], which addresses reasoning like “this does not look like that,” and supports encoded absence) are excluded. This also applies to work on neuronal explanations mentioned earlier. Listing just one or two examples per class is not sufficiently informative and cannot justify claims such as “standard XAI methods”.

[1] Mu, Jesse, and Jacob Andreas. "Compositional explanations of neurons." Advances in Neural Information Processing Systems 33 (2020): 17153-17163.

[2] La Rosa, Biagio, Leilani Gilpin, and Roberto Capobianco. "Towards a fuller understanding of neurons with clustered compositional explanations." Advances in Neural Information Processing Systems 36 (2023): 70333-70354.

[3] Oikarinen, Tuomas, and Tsui-Wei Weng. "Linear explanations for individual neurons." Proceedings of the 41st International Conference on Machine Learning. 2024.

[4] G. Singh and K. -C. Yow, "These do not Look Like Those: An Interpretable Deep Learning Model for Image Recognition," in IEEE Access, vol. 9, pp. 41482-41493, 2021, doi: 10.1109/ACCESS.2021.3064838.

**Questions:**

see weaknesses.

---

> ### Author Response · Authors · 2025-11-25
>
> We thank the reviewer for the constructive feedback and for appreciating our debiasing experiment.
>
> **1.1 Encoded absences have already been studied.** We thank the reviewer for the additional references and fully agree that logical neuron explanations are closely related to our work, as acknowledged in footnote 1. However, our notion of *encoded absences* differs in important ways from the provided references: we require evidence of **active suppression** in the sense of Definition 2.1 (the presence of a concept causes a decrease in activation), rather than simply the absence of activation as used in references [1-2]. We now **discuss this distinction more prominently in Sec. 3**:
> - **Mu & Andreas (2020) [1].** Their method applies Network Dissection (Bau et al., 2017) to identify concepts that *maximally activate* a channel, then searches for logical formulae over these concept masks. As a consequence, only samples with *high positive* activation are considered, and samples that *suppress* a channel (strong negative activation) are effectively excluded from the search space. Moreover, the logical NOT in their work arises from intersections of binary segmentation masks: a concept is treated as part of a logical NOT whenever the neuron simply does not fire in regions where the concept is present. This “mask-based” NOT does not distinguish between “no evidence” and “inhibitory evidence” and thus does not capture the inhibitory notion of encoded absence we formalize. We now highlight this distinction in Sec. 3 (L. 155-167) and empirically in Sec. 5.3, where we show that concepts appearing in NOT expressions from [1] do **not** lead to a strong decrease in activation when inserted into highly activating images.
> - **La Rosa et al. (2023) [2].** This work extends [1] by clustering and considering different activation ranges, but, for ReLU-based models, still focuses on **non-negative** activations. Hence, the same two limitations apply: (i) inhibitory samples are largely ignored, and (ii) NOT remains defined via mask intersections where the neuron fails to activate, rather than via suppression in the sense of Definition 2.1. We clarify this relationship and our different use of “encoded absence” in the revised Sec. 3.
> - **Oikarinen & Weng (2024) [3].** We carefully re-read [3] and did not find an explicit treatment of the property “whose absence causes high activations, or vice versa, whose presence strongly suppresses the activation of a specific internal neuron,” as used in our paper. Their method derives *linear explanations* for ReLU neurons over *positive activation ranges*; accordingly, in their visualizations (e.g., their Fig. 4 and App. Figs. 6-13), activations and weights are positive, suggesting a clear focus on neurons encoding the *presence* of concepts. Their method is complementary to ours and could, in principle, be applied to the most negative activations to obtain linear explanations for encoded absences.
>
> **1.2 & 1.3 The proposed extensions have been widely studied & only the debiasing experiment appears novel.**
>
> As clarified in point 1.1, the provided prior work on logical NOTs does not capture **inhibitory encodings** in the sense of Definition 2.1. In light of this distinction, we believe **our formulation of encoded absence in the context of XAI remains conceptually novel**.
>
> Regarding the methodological components, we agree that the underlying techniques are not new and noted this in the submission (Sec. 4, now L. 247–263). Our contribution is rather a **causal reinterpretation and integration** of these techniques, showing that, together with our formal definition, they enable mainstream XAI methods to uncover inhibitory evidence. Specifically, our contributions are:
>
> 1. A **causal and mechanistic formulation** of encoded absences (Secs. 2.1-2.2), formally distinguishing inhibitory evidence from, e.g., mask-based logical NOTs of [1,2] (a strength listed by Reviewer iv67).
> 2. An analysis showing that mainstream attribution, feature-visualization, and counterfactual methods, in their standard form, tend to **overlook inhibitory relationships** (Secs. 3, 5.1-5.2), a limitation not previously articulated.
> 3. A **unified interpretive framing** explaining why simple variants of these methods (non-target attribution and minimization-based feature visualization) naturally reveal encoded absences in the sense of Definition 2.1.
> 4. Empirical evidence that encoded absences are **prevalent in ImageNet-scale models** (Sec. 5.3) and that explicitly modeling them yields **debiasing improvements** not achievable by presence-only attribution priors (Sec. 5.4).
>
> From this perspective, the debiasing experiment is not the only novelty but the **endpoint** of the broader conceptual contribution: once encoded absences are correctly formalized and exposed, they naturally become available for downstream use. We have streamlined the revised Introduction and Sections 2-4 to make this progression more explicit.

---

> > ### Author Response · Authors · 2025-11-25
> >
> > **2. Missing details and structure.** We thank the reviewer for considering most parts of our paper well written. To clarify the implementation of non-target attributions, we have included the concrete, task-specific computation strategies in Sec. 4 (L. 230-235) and App. 5. We hope these additions make the mechanics of non-target attribution clearer.
> >
> > **3. Overgeneralized statements.** We apologize for the imprecise wording in the original manuscript and thank the reviewer for highlighting this. Our intention was *not* to claim that no existing XAI method can, in principle, capture encoded absences (e.g., we have also listed work on logical NOT (Dukor, 2018) and an intrinsically explainable model (Prabhushankar & AlRegib, 2021) in the initial submission), but rather that the **most commonly used** post-hoc tools, when applied in their standard form, typically overlook them.
> >
> > To make our intent more clear, we replace the term “standard XAI methods” by “mainstream XAI methods” and **explicitly define** this scope in the Introduction and Sec. 3 as: post-hoc attribution methods, feature-visualization techniques, and counterfactual explanations. Further, we soften and qualify our claims. Throughout the paper, we now state that mainstream methods “**struggle to reveal**” encoded absences as defined in Definition 2.1 and when applied “**in their standard form**” (e.g., L. 021-022, 177, 529).
> >
> > We have also streamlined the related-work discussion to focus on methods most closely related to our contributions and added the missing references suggested by the reviewer (including [1,2,4]), ensuring that our claims are properly contextualized.

---

> > ### Comment · Reviewer_1bAo · 2025-11-26
> >
> > Thank you to the authors for their answers. In this message, I focus and provide further clarification below regarding points 1.1 and 1.2, as it appears the original review may not have been sufficiently clear. The other points will be discussed in subsequent messages, if needed.
> >
> > The concern relates to the interpretation of encoded absence and there seems to be a misunderstanding in the discussion. Encoded absence can be defined in two ways:
> >
> > **Def 1**. *If concept X is present, the activation increases; if it is absent, the activation decreases.*
> >
> > This is a **bidirectional condition**. And it is what is shown/proved in **Section 2.2**
> >
> > **Def 2**. *If concept X is present, the activation is low/minimized.*
> >
> > This definition tests only the **presence–suppression relation**. This is aligned with the paper's **definition 2.1**
> >
> > The paper appears to treat these two formulations as equivalent, but they are not.
> > The first definition tests the effect of absence explicitly, while the second measures only whether concept presence is (strongly) associated with low (or minimized) activation.
> >
> > In the current paper, activation minimization (from the literature) is used as a method to capture the first type of relationship. However, by design, it captures only the second phenomenon. This second case has already been explored in prior work by the activation minimization method itself (so it is not novel by itself) and by [2,3], since those studies examine multiple activation levels, including minimum activation. This means they can associate low activation with the presence of a concept, which aligns with Definition 2. Therefore, the concept of "encoded absence" as defined in Def 2 is not novel in literature.
> >
> > Setting aside current implementation limitations, the first definition has been partially explored through the NOT operator, since applying it to high activations effectively represents: "the neuron reaches high activation only if concept X is absent." What is missing is the bidirectional condition. And that condition is what would meaningfully distinguish this work (as the paper tries to support in 2.2) if encoded absence were embedded directly into the feature visualization procedure, enforcing constraints that capture the full bidirectional relationship "the neuron reaches highest activation only if concept X is absent **and** reaches lowest activation when concept X is present".
> >
> > However, the **current proposed workflow** (activation minimization from literature) **does not meet this condition**. Supporting it would require major changes, though such changes could strengthen the contribution. Similar concerns also apply to the attribution component, as highlighted by other reviewers.
> >
> > These points relate to the fact that, as the authors confirmed, the specific techniques used are not novel and have appeared in previous literature. What is novel from their point of view is the "interpretation of their outputs" in terms of encoded absence. However, **if the concept of encoded absence presents weaknesses in terms of novelty (in the case of def 2) or evidential support (in the case of def 1), then the overall contribution is affected** from my point of view.

---

> > > ### Author Response · Authors · 2025-12-03
> > >
> > > We thank the reviewer for the in-depth concerns and for giving us the opportunity to resolve potential misunderstandings.
> > >
> > > ### Clarification on Def 1 vs. Def 2
> > > We thank the reviewer for raising a possible distinction between the bidirectional formulation (their Def 1) and the one-directional formulation (their Def 2). To avoid confusion, we note that the reviewer's Def 1 appears to contain a small directional typo; based on the reviewer’s subsequent explanation ("the neuron reaches high activation only if concept X is absent"), we interpret Def 1 as: activation **increases** when concept $\hat{x}$ is absent and **decreases** when concept $\hat{x}$ is present.
> > >
> > > There are two aspects in the two definitions that benefit from clarification: (i) bidirectionality and (ii) the distinction between "decreased" (as written in their Def 1) and "minimized" (as written in their Def 2).
> > >
> > > **(i) Bidirectionality.** In our Def 2.1, we implicitly (and in the newest revision explicitly) assume a fixed input context $x$, in which a concept $\hat{x}$ can be either present ($[x,C_{\hat{x}}=1]$) or absent ($[x,C_{\hat{x}}=0]$), where $C_{\hat{x}} \in \{0,1\}$ denotes a binary indicator. Def 2.1 states that a neuron encodes the absence of a concept $\hat{x}$ if the presence of $\hat{x}$ reduces the activation: $f([x,C_{\hat{x}}=1]) < f([x,C_{\hat{x}}=0])$. This immediately implies the reverse inequality: $f([x,C_{\hat{x}}=0]) > f([x,C_{\hat{x}}=1]),$ meaning that, in the same context $x$, the **absence of $\hat{x}$ yields higher activation relative to its presence**.
> > > Both statements express the **same ordering** of the two activations with respect to concept presence/absence. Therefore, **under a fixed context**, the "bidirectional" conditions in Def 1 do not describe two different phenomena; it is simply the same comparison stated in both directions. This is also why our constructive existence proof in Sec. 2.2 naturally covers the reviewer’s Def 1. We have made this important detail more explicit in the revision (L. 112-123).
> > >
> > > **(ii) Distinction between *decreased* and *minimized*.** Def 2.1 defines encoded absence through a *decrease* in activation. Feature visualization through minimization, by contrast, identifies the input that achieves the *lowest* activation. This is, however, fully consistent: minimization finds a **subset** of all encoded absences, just as standard activation maximization finds a subset of all encoded presences. We do not claim that minimization recovers the complete inhibitory set, only that it reveals inputs that satisfy Def 2.1 particularly strongly. As noted earlier, methods such as [3] could, in principle, explore intermediate negative activation ranges, but this is **not** done in [3] for ReLU models, where negative activations are collapsed to zero.
> > >
> > > We hope this clarifies that (1) the reviewer’s Def 1 and our Def 2.1 express the same mathematical relationship under a fixed context, and (2) our use of minimization is aligned with our Def 2.1 and does not imply any contradiction. To make these points clearer, we improved the notation in L. 112-123 and added a note on feature visualization through minimization that it can only find a subset of all encoded absences (L. 246).
> > > ### Clarification on novelty and prior work
> > > While activation minimization has been used in prior work, we are still not aware of any established connection to inhibitory evidence (see previous reply for a more detailed answer). For example, Olah et al. (2017) use minimization only to obtain more diverse explanations. References [1,2] rely on logical NOT operators applied to segmentation masks; this **cannot distinguish between an *inhibitory effect* and a *complete lack of functional dependence***, since both cases appear identical in their mask-based logic. As a result, one cannot identify encoded absences, and it would, for example, not be possible to obtain the explanations we demonstrated in Secs. 5.1-5.4. To further verify this empirically, in Tab. 1 of Sec. 5.3, we have measured the inhibitory effects of logical NOT concepts from [1] and see no significant inhibitory effect beyond random patches.
> > > Regarding [3], all explanations for ReLU models operate on post-ReLU activations. In such settings, inhibitory relationships and no dependence on a concept are both mapped to zero, making them **indistinguishable** as well. Further, in Fig. 4 and App. Figs. 6-15 of [3], all examples **start from the top 10% activation range**, and **all concept weights in their linear explanations are positive**. So, even if the full activation range was used, we see no evidence that [3] analyzes or establishes inhibitory effects of the kind studied in our work.
> > >
> > > In contrast, our work (i) formally defines encoded absences as inhibitory evidence, (ii) shows how inhibitory relationships can be made visible using mainstream post-hoc XAI methods, and (iii) demonstrates that these inhibitory signals have downstream utility, e.g., in debiasing.

---

### Meta-Review · Area_Chair_a3Yg · 2026-01-07

**Summary:**

The primary controversy centers on the novelty and distinctness of the proposed "encoded absence" concept compared to existing literature on "Logical NOT" in neurons and negative feature attributions. Reviewer 1bAo strongly contends that the conceptual contribution is marginal, arguing that prior works (e.g., Mu et al., La Rosa et al.) already address absence and that the proposed algorithmic extensions—feature visualization via minimization and non-target attribution—are trivial adaptations. Reviewer iv67 raised significant concerns regarding the causal validity of the method, suggesting that the "least activating patches" might result from distribution shifts or artifacts rather than genuine semantic suppression. Reviewer GVJY initially questioned the differentiation from existing negative attribution methods like Shapley values and "Prototypes and Criticisms," as well as the qualitative nature of the patch analysis.

**Reviewer Concerns:**

The rebuttal resolved Reviewer GVJY’s concerns regarding the distinction from "Prototypes and Criticisms" and the rigor of the patch methodology, resulting in a promised score increase. The authors also attempted to address Reviewer iv67’s critique of distribution shifts by introducing a "random patch" baseline; while this technically mitigated the artifact concern, it failed to bridge the reviewer's perceived "mental leap" regarding causality, leaving that concern outstanding. The most critical outstanding issue is the fundamental deadlock with Reviewer 1bAo regarding novelty: while the authors argued that prior methods only detect "zero activation" whereas their method detects "active suppression", the reviewer rejected this distinction as they feel it is insufficient. AC agrees with reviewer 1bAo and also found the quantitative evaluation of the proposed method lacking in its current state.

**Reviewer Scores:**

Reviewer GVJY will likely increase their score based on the rebuttal clarifications. Reviewer A8Sw was already fully convinced and would maintain their 8 score. Reviewer iv67 actively participated in the discussion and explicitly declined to raise their score despite the new baseline experiments, indicating their reservations about the causal framing are entrenched. Reviewer 1bAo engaged deeply but remained staunchly opposed; further discussion would arguably not change their score, as their rejection is based on a fundamental disagreement about the value of the "inhibitory vs. zero" distinction rather than a misunderstanding of the text.

---

### Decision · Program_Chairs · 2026-01-26

Reject